*Atmos. Chem. Phys.* manuscript:

# Measurement report: Exchange fluxes of HONO over agricultural fields in the North China Plain

Yifei Song[1,3,#], Chaoyang Xue[2,#,*], Yuanyuan Zhang[1,5,*], Pengfei Liu[1,5], Fengxia Bao[2], Xuran Li[4], Yujing Mu[1,5,*]

[1] Research Center for Eco-Environmental Sciences, Chinese Academy of Sciences, Beijing 100085, China

[2] Max Planck Institute for Chemistry, Mainz 55128, Germany

[3] SINOPEC Beijing Research Institute of Chemical Industry, Beijing, 100013, China

[4] Rural Energy and Environment Agency, Ministry of Agriculture and Rural Affairs, Beijing 100125, China

[5] University of Chinese Academy of Sciences, Beijing 100049, China

[#] These authors contributed equally to this work.

**Correspondence:**

Chaoyang Xue (ch.xue@mpic.de)

Yuanyuan Zhang (yyzhang@rcees.ac.cn)

Yujing Mu (yjmu@rcees.ac.cn)

**Abstract**

Nitrous acid (HONO) is a crucial precursor of tropospheric hydroxyl radicals, but its sources are not fully understood. Soil is recognized as an important HONO source, but the lack of measurements of soil-atmosphere HONO exchange flux ($F_{HONO}$) has led to uncertainties in modeling its atmospheric impacts and understanding the reactive nitrogen budget. Herein, we conduct $F_{HONO}$ measurements over agricultural fields under fertilized ($F_{HONO-NP}$) and non-fertilized ($F_{HONO-CK}$) treatments. Our results show that nitrogen fertilizer use causes a remarkable increase in $F_{HONO-NP}$. $F_{HONO-NP}$ exhibits distinct diurnal variations, with an average noontime peak of 152 ng N m$^{-2}$ s$^{-1}$. The average $F_{HONO-NP}$ within three weeks after fertilization is $97.7 \pm 8.6$ ng N m$^{-2}$ s$^{-1}$, around two orders of magnitude higher than before fertilization, revealing the remarkable promotion effect of nitrogen fertilizer on HONO emissions.

We also discuss other factors influencing soil HONO emissions, such as meteorological parameters and soil properties/nutrients. Additionally, we estimate the HONO emission factor of $0.68 \pm 0.07\%$ relative to the applied nitrogen during the whole growing season of summer maize. Accordingly, the fertilizer-induced soil HONO emission is estimated to be 22.3 and 60.8 Gg N yr$^{-1}$ in the North China Plain (NCP) and mainland China, respectively, representing a significant reactive nitrogen source. Furthermore, our observations reveal that soil emissions sustain a high level of daytime HONO, enhancing the atmospheric oxidizing capacity and aggravating O$_3$ pollution in the NCP. Our results indicate that to mitigate regional air pollution effectively, and future policies should consider reactive nitrogen emissions from agricultural soils.

**Keywords**

HONO; Agriculture fields; Nitrogen fertilizer; North China Plain; Reactive nitrogen budget; Air pollution

**1 Introduction**

Hydroxyl radical (OH) is the major oxidant in the troposphere, which can oxidize primary pollutants (volatile organic compounds, NO$_X$, SO$_2$, etc.), with the formation of secondary pollutants (aerosols, O$_3$, etc.). It also determines the lifetime of some greenhouse gases like methane, which affects global climate (Seinfeld and Pandis, 2016). It is, therefore, necessary to understand the OH formation path. Nitrous acid

(HONO) is an important primary OH source, with a contribution of 20–90% to primary OH production in the lower troposphere (Kim et al., 2014; Song et al., 2022a; Tan et al., 2017; Tan et al., 2018; Xue et al., 2020). However, the source of HONO is still incompletely understood, especially during daytime (Jia et al., 2020; Kleffmann et al., 2007; Li et al., 2018; Xue et al., 2022a). Recently, unexpectedly high HONO concentrations up to ppbv level were observed during daytime, suggesting strong daytime missing sources of 0.1–4.9 ppbv h$^{-1}$ in urban and rural areas (Acker et al., 2006; Kleffmann and Gavriloaiei, 2005; Li et al., 2012; Li et al., 2014). In addition, several previous studies have reported significant gradients in vertical HONO distribution, indicating a strong HONO source at the ground surface (Kleffmann et al., 2007; VandenBoer et al., 2013; Wong et al., 2012; Zhang et al., 2009). Among daytime HONO sources, the ground-derived HONO sources mainly include (1) photo-enhanced heterogeneous reaction of $NO_2$ on the soil surface (George et al., 2005; Han et al., 2016; Stemmler et al., 2006; Stemmler et al., 2007), (2) photolysis of adsorbed nitric acid (Zhou et al., 2011) on the ground surface, (3) release of adsorbed HONO from strong acid (HCl, $HNO_3$, etc.) displacement (VandenBoer et al., 2015) and (4) soil emissions from biogenic progress, which is considered as an important daytime HONO source in agricultural areas and has been proved by many laboratory experiments and several field flux measurements (Oswald et al., 2013; Su et al., 2011; Tang et al., 2019; Tang et al., 2020; Xue et al., 2019a). Under laboratory conditions, Oswald et al. (2013) found that soil mineral nitrogen is substantially associated with HONO emissions, which suggests nitrogen fertilizer use can greatly enhance the potential of soil HONO emissions. In our recent study, elevated levels of HONO concentration and HONO-to-$NO_2$ ratios were observed after fertilization at an agricultural site, implying that fertilized fields released a great amount of HONO (Xue et al., 2021). Therefore, conducting direct flux measurements is necessary to study the characteristics and the corresponding atmospheric impacts of soil HONO.

Flux measurement can provide direct evidence about the production and/or deposition of HONO on the ground surface (von der Heyden et al., 2022; Xue et al., 2022b). The aerodynamic gradient (AG) and relaxed eddy accumulation (REA) methods have been developed and applied to HONO flux measurements in recent years, providing a good option to measure HONO flux (Laufs et al., 2017; Sörgel et al., 2015; von der Heyden et al., 2022; Zhou et al., 2011). As shown in Table 1, Laufs et al. (2017) and Sörgel et al. (2015) measured HONO fluxes using the AG method above bare soil, different crops, and forests, and they found the flux was mainly contributed by $NO_2$-related photosensitized reactions. Ren et

al. (2011), Zhou et al. (2011), and von der Heyden et al. (2022) draw similar conclusions using the REA method above agricultural fields, forests, and grassland, respectively, and the maximum HONO fluxes in these studies were all less than 20 ng N m$^{-2}$ s$^{-1}$. Apart from the above meteorological methods, the dynamic chamber method uses ambient air to flush the chamber to avoid the formation of water film, allowing the determination of HONO exchange fluxes between soil and atmosphere (Tang et al., 2019; Xue et al., 2019a). Different from the above measurements, Xue et al. (2019a) and Xue et al. (2022b) observed extremely high levels of HONO fluxes under "over-fertilized" conditions, with the maximum fluxes being 1515 and 348 ng N m$^{-2}$ s$^{-1}$, respectively, which approach even exceed the values in the laboratory experiments (Oswald et al., 2013; Wang et al., 2021). Besides, the mechanism of soil HONO emissions is still incompletely understood, e.g., the dominant source of soil nitrite (NO$_2^-$) (Bhattarai et al., 2021; Ermel et al., 2018; Oswald et al., 2013; Scharko et al., 2015; Song et al., 2023a; Wu et al., 2019), the chemical-physical transformation of soil NO$_2^-$ to gas-phase HONO (Bao et al., 2022; Kim and Or, 2019; Oswald et al., 2013; Su et al., 2011; Xue et al., 2022b). Moreover, available flux measurements are still limited; most were conducted over a short period of normally less than one month. A systematic and relatively longer measurement covering a whole growing season of a crop is lacking, resulting in limitations in estimating the HONO emission factor (EF$_{HONO}$) as well as the understanding of the reactive nitrogen budget at an annual scale (Xue et al., 2022b).

With the reduction in reactive nitrogen emissions from anthropogenic combustion processes, natural and agricultural field emissions are becoming increasingly important. About one-third of the world's nitrogen fertilizer is consumed in China, indicating a strong potential for reactive nitrogen emissions. Moreover, with the agricultural intensification, more and more farmland implemented mechanization operations, leading to changes in fertilizer application methods. Currently, two fertilizer application methods are used in China: deep fertilization (DF) during machine sowing and spreading fertilizer (SF) on the soil surface (Nkebiwe et al., 2016; Pan et al., 2017), and the former one is becoming popular used in recent years. Our recent study observed high soil HONO emissions under SF conditions (Xue et al., 2022b). Therefore, it is necessary to conduct measurements under DF conditions, considering that the emissions may change as fertilizer application method.

In this study, soil-atmosphere HONO exchange fluxes were measured by an open-top dynamic chamber (OTDC) system during the whole growing season of summer maize in the North China Plain (NCP).

HONO fluxes from soils with several treatments were determined in parallel, which enables the understanding of the influencing factors of HONO emissions and the discussion of potential emission reduction strategies, etc. In addition, for the first time, the cumulative emissions and emission factors of HONO from the agricultural soils were calculated based on our flux measurements crossing a whole growing season, benefitting the estimation of yearly soil HONO emissions at a national scale assessment of their atmospheric impacts.

## 2 Methods

### 2.1 Study site

The field flux measurement was conducted at the Station of Rural Environmental, Research Center for Eco-Environmental Sciences (SRE-RCEES, 38°71′N, 115°15′E), located in Wangdu County, Hebei Province, China. The station is surrounded by vast agricultural fields, with winter wheat and summer maize rotation. More detailed descriptions of the measurement site can be found in our previous studies (Liu et al., 2017; Song et al., 2022a; Song et al., 2022b).

During the summer maize season, two different treatments were designed in the experiment fields: CK (control, normal flood irrigation but no fertilization for decades) and NP (fertilizer deep placement and normal flood irrigation, same as local farmers). In the experimental fields, summer maize was sown on June 17, 2021, and harvested on September 27, 2021. Only one fertilizer application event was conducted during the maize season. A typical-used compound fertilizer (N: $P_2O_5$: $K_2O$ = 28%: 6%: 6%) was mechanized and buried to a depth of 8–10 cm when sowing maize seeds. The fertilizer application rate (FAR) of the NP treatments was 300 kg N ha$^{-1}$, the typical amount local farmers used. According to the fertilization event and the characteristics of HONO flux, the whole observation was divided into three periods: (1) Pre-fertilization period (PFP, before June 18); (2) High HONO emission period (HEP, from June 18 to July 10); (3) Low HONO emission period (LEP, after July 10).

### 2.2 Flux measurements

HONO exchange fluxes were measured by an OTDC system, which is updated based on our previous design (Xue et al., 2019a) (I.D. of 32 cm, 80 cm in height). Eight chambers (six experimental chambers, Exp-chambers, and two reference chambers, Ref-chambers) were divided into two groups (NP and CK)

to obtain HONO flux from soils with different treatments (see Section 2.1). As shown in Figure 1, each group contains three replicated Exp-chambers and one Ref-chamber that are flushed by the same air pumped by diaphragm pumps (N838KNE, Germany) from the top of the metal sample tube (I.D. of 4 cm, 2 m in height, with the inner wall coated with Teflon film). The layout and construction of the OTDC system and other equipment are shown in Figure 1. Note that both fluxes from the NP and CK plots include heterogeneous HONO formation on the ground surface, which leads to an overestimation of soil HONO flux (Xue et al., 2019a). By comparing between flux from the NP and CK groups, we can distinguish the relative importance of soil emissions and $NO_2$ heterogeneous reactions and determine the net effect of fertilizer use. However, the current OTDC still has some problems, such as drying of the soil by flushing gas from the bottom, the greenhouse effect in the chambers, and the formation of HONO originating from soil $NO_X$ emissions, etc., which will result in the overestimation of the HONO flux. These issues should be considered in the future.

HONO was continuously sampled by stripping coils with the absorption solution of ultrapure water. The sampling interval was normally set at 12 h for all four chambers from the CK group during the whole measurement period. For the NP group, the sampling interval was set at 2 h for one Exp-chamber and one Ref-chamber during PFP and HEP, and 12 h for the other two Exp-chambers in parallel. The 2-h interval measurements can provide detailed information about the diurnal variation. All samples were timely analyzed by an ion chromatography system (IC6200, WAYEE, China) (Xue et al., 2019b).

The soil HONO exchange flux ($F_{HONO}$, ng N $m^{-2}$ $s^{-1}$) can be obtained by the difference of HONO concentrations in the Exp-chambers and Ref-chamber (Xue et al., 2019a):

$$F_{HONO} = \frac{(HONO_{exp} - HONO_{ref}) \times F_{flush} \times M_N \times P}{R \times T \times S} \qquad (eq–1)$$

Where $HONO_{exp}$, $HONO_{ref}$, $F_{flush}$, $M_N$, P, R, T, and S are the HONO concentrations (ppbv) in the Exp-chambers and Ref-chambers, the flushing flow ($3.33 \times 10^{-4}$ $m^3$ $s^{-1}$), the molar mass of N (ng $mol^{-1}$), the atmospheric pressure (Pa), the ideal gas constant (8.314 J $mol^{-1}$ $K^{-1}$), the atmospheric thermodynamic temperature (K), and the area of the soil covered by the chamber ($m^2$), respectively.

The emission factor of HONO ($EF_{HONO}$, %) relative to the amount of applied nitrogen can be calculated based on the following formula:

$$EF_{HONO} = \frac{E_{NP} - E_{CK}}{FAR} \times 100\% \qquad (eq–2)$$

where $E_{NP}$ and $E_{CK}$ are the cumulative HONO emissions (kg N ha$^{-1}$) from the fertilized plots and the control plots, respectively. FAR is 300 kg N ha$^{-1}$.

## 2.3 Measurements of meteorology and soil characteristics

Meteorological parameters, including temperature, relative humidity (RH), pressure, wind speed (WS), wind direction (WD), precipitation, and solar radiation (SR), were recorded by an auto weather station (Vaisala WXT520, Finland). The photolysis frequency of NO$_2$ (J(NO$_2$)) was measured by a 2-$\pi$ J(NO$_2$) filter radiometer (MetCon, Germany). However, J(NO$_2$) was not measured during August 18–27. Instead, it was estimated via a high correlation between SR and J(NO$_2$) (Figure S1, J(NO$_2$) = -5.20×10$^{-9}$ m$^4$ W$^{-2}$ s$^{-2}$ × SR + 1.17×10$^{-5}$ m$^2$ W$^{-1}$ s$^{-1}$ × SR, $R^2$ = 0.91).

The soil temperature at ~5 cm depth was auto-monitored by a sensor. The moisture of topsoil (0–5 cm) was expressed as water-filled pore space (WFPS), which was calculated by dividing the volumetric water content by the total soil porosity. The volumetric water content was measured twice a day (9:00 and 21:00 local time (LT)) by a soil humidity sensor (Stevens Hydra Probe Ⅱ, USA) when collecting HONO samples, and total soil porosity was calculated according to the relationship: soil porosity = (1 – soil bulk density / 2.65), assuming a particle density of 2.65 g cm$^{-3}$, which is the most commonly used value for mineral soils (Linn and Doran, 1984). The topsoil samples were taken once a week by a ring sampler (5 cm diameter × 5 cm height) and the topsoil bulk density was determined gravimetrically by oven drying at 105 °C for 12 h.

To analyze the soil NH$_4^+$-N and NO$_3^-$-N concentrations, soil samples were collected every three days for both NP and CK fields. Each sample was collected in four points and homogeneously mixed. 20 g of the mixed soil was extracted with 100 ml of 1 mol L$^{-1}$ KCl solution, shaken in a rotary shaker (140 r min$^{-1}$) for 1 h, and filtered into sampling bottles. Samples were frozen until analyzed by a colorimetric continuous flow analyzer (Seal Analytical AutoAnalyzer 3, USA).

## 2.4 Data analysis

Mean tests, variance tests, and correlation analysis about HONO flux and other parameters were performed using the statistical software SPSS Statistic 24 (SPSS Inc., Chicago, USA). The graphing

software Origin 2018 (Origin Lab Corporation, Northampton, MA, USA) and ArcGIS 10.5 (ESRI Inc., California, USA) created figures.

## 3 Results and discussion

### 3.1 The variation of key meteorological and soil parameters

The variations of key meteorology (air temperature, RH, pressure, rainfall, and $J(NO_2)$) and soil parameters (soil $NH_4^+$-N and $NO_3^-$-N concentrations, temperature, and WFPS) during the measurement period are shown in Figure 2 and Figure S2, respectively. During the whole maize growing season (from June to September), average air temperature, RH, and soil temperature were 24.8°C, 72.3%, and 23.7°C, respectively. Except for the value less than $1.0 \times 10^{-4}$ s$^{-1}$, the average $J(NO_2)$ during the campaign was $3.12 \times 10^{-3}$ s$^{-1}$. The accumulative rainfall was 514 mm during this period, significantly higher than that in previous years (195–302 mm in the summer maize season during 2008–2011) (Zhang et al., 2014). The measured soil WFPS ranged from 33% to 82% and quickly increased following irrigation or precipitation, with a mean value of 59.1%. In the NP plots, the soil $NH_4^+$-N and $NO_3^-$-N concentrations increased significantly after applying nitrogen fertilizer (with averages of 48.0 and 112 mg kg$^{-1}$ within 20 days after fertilization). In contrast, they remained at much lower levels throughout the whole maize season in the CK plots (with averages of 4.63 and 4.45 mg kg$^{-1}$).

### 3.2 Characteristics of HONO flux

Figure 3 shows the time series of HONO fluxes at a 12h-interval (daytime: 9:00–21:00, nighttime: 21:00–9:00) from the NP and CK plots ($F_{HONO-NP}$ and $F_{HONO-CK}$, average of three in-parallel duplications) during the whole maize season of 2021. HONO emission after fertilization showed a distinctly diurnal variation, that is, it was significantly higher in the daytime than at night, which will be discussed in the next section. During the PFP period, the average $F_{HONO-NP}$ and $F_{HONO-CK}$ were $0.55 \pm 0.35$ and $-0.51 \pm 0.13$ ng N m$^{-2}$ s$^{-1}$ from the NP and CK plots, respectively (Table S1). The higher level of $F_{HONO-NP}$ than $F_{HONO-CK}$ might be ascribed to the residual effect of fertilizer in the NP plots, considering that CK plots have not been fertilized for years. After fertilization, as shown in Figure 3, $F_{HONO-NP}$ gradually increased and peaked on the fifteenth day. The measured $F_{HONO-NP}$ then trended downward but still maintained at a high level of 100 ng N m$^{-2}$ s$^{-1}$ within 3 weeks after fertilization. In contrast, $F_{HONO-CK}$ always fluctuated around zero.

During HEP, the average $F_{HONO-CK}$ value was $-0.36 \pm 0.04$ ng N m$^{-2}$ s$^{-1}$, which is similar to the PFP period (Table S1). In comparison, the mean value of $F_{HONO-NP}$ during HEP was $97.7 \pm 8.6$ ng N m$^{-2}$ s$^{-1}$, revealing the large potential of fertilized soils in HONO emissions. A heavy rainstorm (with a cumulative rainfall of more than 100 mm) fell during 11–13 July, significantly limiting HONO release from the soil. During LEP, the average $F_{HONO-NP}$ and $F_{HONO-CK}$ were 1.03 and $-2.88$ ng N m$^{-2}$ s$^{-1}$, respectively. Overall, except the data from NP plots during HEP, the other fluxes (including fluxes from CK plots during all periods and fluxes from NP plots during PFP and LEP) are similar to observations at other sites with no nitrogen fertilization application, such as grass (von der Heyden et al., 2022) or forest (Ramsay et al., 2018; Sörgel et al., 2015; Zhou et al., 2011) regions.

During the whole maize growing season, negative values of $F_{HONO-CK}$ were frequently observed at night, accounting for ~70% of the total investigated data. Numerous studies on HONO flux measurement from unfertilized fields also reported this phenomenon, which is ascribed to nocturnal HONO deposition (Laufs et al., 2017; Ren et al., 2011; Tang et al., 2020), indicating the complexity of the role of the ground surface in nocturnal HONO production and deposition (Ramsay et al., 2018; VandenBoer et al., 2013; VandenBoer et al., 2015; von der Heyden et al., 2022). Nevertheless, it is worth noting that the high soil moisture after irrigation for the unfertilized CK plots may reduce the conversion from liquid phase HNO$_2$ to gas phase HONO in soils and accelerate the absorption of atmospheric HONO at the soil surface, thereby contributing to the negative HONO fluxes.

**3.3 Diurnal variations of $F_{HONO-NP}$**

Figure 4 shows the time series of $F_{HONO-NP}$ with an interval of 2 h, which shows similar levels and trends to 12h interval measurements. Diurnal variations of HONO$_{exp}$, HONO$_{ref}$, and $F_{HONO-NP}$ from the NP plot during PFP and HEP are shown in Figure 5. During PFP, $F_{HONO-NP}$ exhibited a distinct diurnal variation (Figure 5A), with a maximum of 0.38 ng N m$^{-2}$ s$^{-1}$ at noon and a minimum of 0.07 ng N m$^{-2}$ s$^{-1}$ in the early morning (Figure 5A).

During HEP, HONO$_{exp}$ increased by 1–2 orders of magnitude compared to that during the PFP period. As shown in Figure 5B, the $F_{HONO-NP}$ during HEP also showed "bell-shaped" diurnal variations, but with a much higher peak of 152 ng N m$^{-2}$ s$^{-1}$ at noon and a minimum of 40 ng N m$^{-2}$ s$^{-1}$ in the early morning. Similar diurnal variation trends of HONO flux have been reported in previous studies (Tang et al., 2019;

Tang et al., 2020; von der Heyden et al., 2022; Xue et al., 2019a; Zhou et al., 2011), while some studies also found morning peaks of the diurnal HONO flux (Laufs et al., 2017; Meng et al., 2022; Ren et al., 2011; von der Heyden et al., 2022). Laufs et al. (2017) and Ren et al. (2011) also conducted flux measurements in farmland, and they found high correlations between HONO flux and the product of $NO_2$ concentrations ($[NO_2]$) and $J(NO_2)$ or solar radiation. A similar good correlation was found during the

PFP period of our present study (Figure S3A, $R^2 = 0.89$). These findings suggest that photosensitized heterogeneous reactions of $NO_2$ on the soil surfaces may be the main sources of the observed HONO flux under typical non-fertilization conditions. However, after fertilizer application, HONO flux from the NP plot showed a weaker correlation with the product $[NO_2] \times J(NO_2)$ during HEP (Figure S3B, $R^2 = 0.43$). Moreover, the fluxes from the CK plots (no fertilization, representative for $NO_y$-to-HONO conversion on

the ground surface, see Method) are about 2 orders of magnitude lower than those from the NP plots during HEP. Therefore, the $NO_2$ or other $NO_y$ reactions are not the main drivers of the observed HONO fluxes during HEP.

**3.4 Comparison with previous flux measurements**

The observed HONO flux, measurement methods, and FAR in previous field measurements are shown in

Table 1. With the increase of FAR, the emission of nitrogenous gas tends to show nonlinear growth (Grant et al., 2006; Ma et al., 2010; McSwiney and Robertson, 2005; Shcherbak et al., 2014; Xue et al., 2022b). Except for previous measurements at this agricultural site (Tang et al., 2019; Xue et al., 2022b; Xue et al., 2019a), fluxes from all other measurements are much lower than that in this study, which should be due to the difference in FAR. In this study, the maximum HONO flux at 2h-interval was 372 ng N m$^{-2}$ s$^{-1}$,

which was in the range of the maximums of HONO flux in previous field measurements (up to1515 ng N m$^{-2}$ s$^{-1}$ ) (Tang et al., 2019; Tang et al., 2020; Xue et al., 2019a). In three previous studies in the summer maize field at the SRE-RCEES station, the peaks occurred within one week after fertilization, but their levels were variable depending on the FAR (Tang et al., 2019; Xue et al., 2022b; Xue et al., 2019a). However, the peak of HONO flux occurred approximately two weeks after fertilization during this

campaign. As mentioned in Section 2.1, fertilizers were buried to a depth of 8–10 cm in this study, while they were spread on the soil surface in previous studies (Tang et al., 2019; Xue et al., 2022b; Xue et al., 2019a). Therefore, the relatively late occurrence of the HONO flux peak is probably caused by the slow

dissolution due to the deep placement of nitrogen fertilizer. In addition to the impact on the occurrence time of peak flux, fertilizer application methods also lead to differences in levels of HONO emission fluxes. Compared with our study of fertilizer deep placement, Xue et al. (2019a) observed a peak HONO emission of 1515 ng N $m^{-2}$ $s^{-1}$ under the SF method, which is 4 times the peak in this study (372 ng N $m^{-2}$ $s^{-1}$), although their fertilizer rates are similar (300 kg N $ha^{-1}$ in this study and 330 kg N $ha^{-1}$ in Xue et al. (2019a)). Our recent study found high exponential correlations between maximum HONO flux ($F_{max}$) and FAR, that is, $F_{max} = 4.7 \times exp$ (FAR / 57.3) + 15.3 ($R^2 = 0.998$, SF method, 0–350 kg N $ha^{-1}$) (Xue et al., 2022b). With the FAR used in this study (300 kg N $ha^{-1}$), the predicted $F_{max}$ is ~890 ng N $m^{-2}$ $s^{-1}$, much higher than the observation, indicating that compared to the SF method, the DF method can reduce soil HONO emissions, possibly through minimizing the mineral nitrogen content in topsoil (Ke et al., 2018; Liu et al., 2015; Weber et al., 2015). Therefore, the DF method is expected to be able to help reduce reactive nitrogen emissions from agricultural activities but needs more future systematic assessments.

**3.5 Possible influencing factors**

3.5.1 Rainfall and soil moisture

Soil $NO_2^-$ generally originates from nitrification and/or denitrification processes (Bhattarai et al., 2021; Scharko et al., 2015). Soil $NO_2^-$ (aq) can combine with $H^+$ (aq) and form $HNO_2$ (aq), which is an acid-base process. $HNO_2$ (aq) can be released into the atmosphere as HONO through liquid-gas partitioning (Bao et al., 2022; Su et al., 2011). As reported by Bao et al. (2022), equilibrium HONO concentrations (HONO*) directly affect soil HONO emissions. Rainfalls dilute soil $NO_2^-$ concentration, and thus HONO* decreases, inhibiting HONO emissions. Besides, high soil moisture after rainfalls will decrease gas diffusion in the soil profile, which could not only mitigate HONO transportation in the soil but also inhibit soil nitrification, the most important microbial process for HONO production (Scharko et al., 2015), by decreasing $O_2$ supplement, thus reducing HONO emissions (Wu et al., 2019). On the five rainy days (rainfall > 1 mm) during HEP (June 23 and 27, July 1, 3, and 7), the soil WFPS immediately increased after rainfalls, and the observed soil HONO fluxes decreased sharply as expected (Figure S4). Apart from June 23, the daytime HONO fluxes on the second day decreased to ~70% of the flux on the previous day. This finding suggests that rainfall significantly reduces soil HONO emissions, but a new

equilibrium can be reached shortly after the rain, although the soil water content was still high (Bao et al., 2022; Wang et al., 2021; Wu et al., 2019).

### 3.5.2 Temperature, atmospheric humidity, and solar irradiance

Atmospheric temperature, humidity, and light may also affect HONO soil emissions, given that they have similar or opposite diurnal variations to soil HONO emissions (Figure 5). Atmospheric temperature varies

with soil temperature and can represent the topsoil temperature. At high temperatures, the nitrification process is more active in producing $NO_2^-$ (Tourna et al., 2008), surface water evaporates faster, and thus more HONO is released (Su et al., 2011). Therefore, higher temperatures could promote soil HONO emission, as reported by laboratory studies (Oswald et al., 2013; Xue et al., 2022b), which also helps to explain the observed diurnal profiles of HONO flux.

This study found that the air temperature positively correlated with HONO fluxes during PFP (Figure 6A, $R = 0.73$) and HEP (Figure 6B, $R = 0.93$). In contrast, HONO fluxes behave oppositely with air RH, with high correlation coefficients of 0.74 and 0.93 during PFP and HEP, respectively (Figure 6C–D). Higher RH could inhibit the water evaporation of the topsoil and may also increase topsoil content, limiting the release of soil $NO_2^-$ to the atmosphere in the form of HONO (Bao et al., 2022; Su et al., 2011; Weber et

al., 2015; Xue et al., 2022b).

Additionally, solar radiation seems to be a factor that favors soil HONO emissions, given that a positive correlation ($R > 0.8$) was found between diurnal HONO flux and $J(NO_2)$ (Figures 6E–F). Before fertilization, soil HONO emissions may originate from light-related processes, such as the photosensitive heterogeneous reactions of $NO_2$ on the soil surfaces, the photolysis of nitrate on the soil surface, etc.

(Laufs et al., 2017; Ren et al., 2011; Stemmler et al., 2006; von der Heyden et al., 2022; Zhou et al., 2011). However, as discussed before, the observed fluxes after fertilization were mainly from microbial processes rather than surface $NO_y$-to-HONO reactions. Therefore, the strong correlations between $J(NO_2)$ and HONO flux may be because solar irradiance could warm and dry the topsoil (Tang et al., 2019). Hence, we suspect that solar radiation indirectly affects soil HONO emissions by affecting the

temperature of the topsoil. This is also consistent with our previous laboratory experiments demonstrating no enhancement effect of radiation on soil HONO emissions (Xue et al., 2022b).

### 3.6 Atmospheric impacts and implications

### 3.6.1 Impacts on daytime HONO budget

Strong daytime unknown HONO sources are commonly observed in the summer NCP, which may be related to soil emissions (Liu et al., 2019; Song et al., 2022a; Xue et al., 2021). Therefore, it is necessary to study whether the soil HONO emissions can explain the unknown source strengths. On average, the observed HONO flux was around 43–153 ng N m$^{-2}$ s$^{-1}$ (Figure 5B) in the daytime during HEP. The flux can explain unknown HONO strength of 2.7–9.6 ppb h$^{-1}$ when assuming a mixing layer height of 100 m (Song et al., 2022a; Su et al., 2011; Xue et al., 2022a), which could cover the reported unknown HONO source (1.6–4.3 ppb h$^{-1}$) in the agricultural regions (Liu et al., 2019; Su et al., 2008; Xue et al., 2021). Hence, the HONO emission from fertilized soil acts as an important and even dominant source to explain the missing daytime HONO source, suggesting the potential impact of soil HONO emissions on regional air pollution and revealing the necessity of implementing soil HONO emissions in regional chemistry-transport models.

### 3.6.2 Implication on the reactive nitrogen budget

Flux measurements throughout the whole growing season allow the estimation of cumulative emissions and emission factors, which helps to study the impact of fertilizer-derived HONO emissions on the reactive nitrogen budget. This is becoming more and more important relative to the decreasing anthropogenic emissions in China. However, to the best of our knowledge, there is no report on HONO cumulative emissions and emission factors from nitrogen fertilizer applied to agriculture fields. During HEP, as shown in Figure 3, the flux shows a distinct diurnal variation and a varying daily peak relative to days after fertilization. Therefore, we conducted intensive flux measurements with a 2-h resolution to capture the diurnal variation and establish the average flux. The latter is subsequently used in calculating the cumulative HONO emissions during HEP. And the missing data during this period were filled in by time interpolation. In contrast, fluxes during LEP show no significant daily variation. Hence, the average of measurements with 12-h resolution can effectively characterize the emission level during LEP, and the resulting average flux is further used in calculating the cumulative HONO emissions during this period. Accordingly, we obtained the $E_{NP}$ and $E_{CK}$ during the whole growing season of summer maize of $1.95 \pm 0.21$ and $-0.09 \pm 0.03$ kg N ha$^{-1}$, respectively, and the $EF_{HONO}$ of $0.68 \pm 0.07\%$ relative to the total applied nitrogen. The obtained $EF_{HONO}$ is even at a similar level to $EF_{NO}$ (0.24–0.82%) and $EF_{N2O}$ (1.1–3.8%) as

observed from the maize fields at the same site (Tian et al., 2017b; Tian et al., 2017a; Zhang et al., 2014), indicating the non-negligible contribution of soil HONO emission in soil nitrogen loss (Wang et al., 2023). On a national scale, although the total fertilizer application amount has shown a downward trend after 2015, the current fertilizer application amount is still higher than that in 1978 by a factor of 5 (Figure 7A).

Assuming nitrogen accounts for 22% of the compound fertilizer (N: $P_2O_5$: $K_2O$ = 28%: 6%: 6%), the total applied nitrogen (TN) and FAR in each province in 2021 were listed in Table S2 (data source: the China Statistical Yearbooks 2022, http://www.stats.gov.cn/sj/ndsj/2022/indexch.htm, last access: October 6, 2023). The FARs in each province are in the range of 40–250 kg N ha$^{-1}$. The relationship between the $F_{max}$ under SF condition ($F_{max-SF}$) and FAR was fitted based on our previous study, in which flux

measurements were conducted in different FAR conditions (Xue et al., 2022b) (Figure S5, $F_{max-SF}$ = 29.54 × exp (FAR / 98.04 – 20.19, FAR range: 0–250 kg N ha$^{-1}$). The reduction proportion of DF to SF method was calculated by substituting the FAR in this study (300 kg N ha$^{-1}$) into the above formula and comparing it with the measured maximum flux (372 ng N m$^{-2}$ s$^{-1}$). Accordingly, the $F_{max}$ under DF condition ($F_{max-DF}$) is ~61% of $F_{max-SF}$ under the same FAR, indicating that the DF method can reduce 39% HONO

emissions compared to the SF method. Besides, we assume that the HONO cumulation emission is proportional to $F_{max}$. Therefore, the $EF_{HONO}$ (%) at different FAR levels can be calculated by:

$$F_{max-DF} = F_{max-SF} \times 0.61 \tag{eq–3}$$

$$EF_{HONO} = 0.68 \times \frac{F_{max-DF}}{372} \times \frac{300}{FAR} \tag{eq–4}$$

Note that 0.68 and 372 represent the aforementioned EF, measured $F_{max-DF}$ at a FAR of 300 kg N ha$^{-1}$,

respectively. Based on (eq-3) and (eq-4), the $EF_{HONO}$ is in the range of 0.15–0.5% at a FAR range of 40– 250 kg N ha$^{-1}$ (Table S2). Figure 7B shows the regional fertilizer-induced soil HONO emissions in each province, and the national soil HONO emission is around 60.8 Gg N yr$^{-1}$. Compared to a recent study (Wu et al., 2022), in which HONO emission of 0.52 Tg N yr$^{-1}$ on a national scale is estimated, our estimate is much lower. The large discrepancy could be due to the differences between those two studies.

1) Soil area difference: this study only considers the agricultural soils with nitrogen fertilizer application, $1.7 \times 10^6$ km$^2$ (data source: the China Statistical Yearbooks 2022, http://www.stats.gov.cn/sj/ndsj/2022/indexch.htm, last access: October 6, 2023), which is only one-sixth of the national land area.

2) Time scale difference: this study aims to quantify the fertilizer effect, and the significant positive flux is only observed within ~1 month after fertilizer application, while Wu et al. (2022) estimate annual emissions.

3) Bi-directional effect: soil-atmosphere is a bi-directional process, e.g., the soil surface can release or absorb HONO (Bao et al., 2022; VandenBoer et al., 2013), which can also be verified by the occurrence of positive and negative fluxes in this and other studies (Table 1). Current flux measurements measured the net flux of the bi-directional soil-atmosphere exchange, while the calculated estimate only considers the soil-to-atmosphere process (Weber et al., 2015; Wu et al., 2022).

Considering that the NCP region consumes a large portion of fertilizer in China and suffers severe $O_3$ pollution, soil HONO emissions need to be a particular concern in this region. As shown in Figure 7, the TN and the corresponding soil HONO emissions were 8.0 Tg N and 22.3 Gg N yr$^{-1}$ in the NCP in 2021, respectively. The soil HONO emission is more than 10% of the annual regional soil $NO_X$ emissions in the NCP ($0.18 \pm 0.01$ Tg N yr$^{-1}$) (Lu et al., 2021), indicating that fertilized soils are important primary sources of both HONO and $NO_X$, exceeding other primary HONO emissions, such as livestock dung (Maljanen et al., 2016; Zhang et al., 2023), and affecting regional air quality in the NCP. It may also indicate that soil HONO emissions should be considered in environmental policies to mitigate regional air pollution.

3.6.3 Implication on regional $O_3$ pollution

HONO emitted from fertilized soil can enhance regional oxidation capacity, leading to $O_3$ formation. Fertilized soils could also emit $NO_X$, which may also aggravate $O_3$ pollution. For instance, Xue et al. (2021) conducted field measurements before and after fertilization periods and found that the average daytime $O_3$ concentration increased from 50 ppbv during the no-fertilization period to 70 ppbv during the intensive fertilization period. Wang et al. (2021) conducted model simulations with consideration of laboratory-derived soil HONO emissions and found that soil HONO emissions enhance the daytime OH concentration by 41% and $O_3$ by 8% in the NCP. Moreover, Wu et al. (2022) estimated regional soil HONO emissions enhanced daytime OH concentrations by 10–60% and daytime $O_3$ concentrations by 0.5–1.5 ppb in Shanghai, China. A recent estimation revealed that daily maximum 8-h average $O_3$ enhancement from soil HONO and $NO_X$ emissions of 8.0 ppbv over the NCP and 5.5 ppbv over China in

June–July 2019 (Tan et al., 2023). In this study, our fluxes measurements cover the non-fertilization and fertilization periods, which allows an estimation of the impact of soil HONO and $NO_X$ emissions on atmospheric composition, such as $O_3$ pollution. To do that, we use "$O_3+NO_2$" to represent the total oxidant concentrations ($O_X$) (Tang et al., 2009). Figures S6 and 8 show the time series and diurnal variation of $O_3$, $NO_2$, and $O_X$ concentrations during PFP, HEP, and LEP, respectively. Compared with the PFP and LEP, the $O_3$ and $NO_2$ concentrations were higher during HEP by factors of 1.35–1.67 and 1.17–1.51, respectively. The averaged $O_X$ concentrations were $68.4 \pm 29.5$ ppbv during HEP, 16.5 ppbv (31.8%) and 26.7 ppbv (64.1%) higher than those during PFP and LEP, respectively. The results demonstrate that fertilization can significantly affect regional $O_3$ formation in the NCP. Considering that the NCP region is a hot spot of $O_3$ pollution in recent years (Ma et al., 2021; Wang et al., 2017; Wang et al., 2020), the impacts of fertilizer-derived HONO emissions should be considered in terms of diagnosing $O_3$ pollution. Therefore, although nitrogen fertilizer use is necessary to enhance crop yield, it can result in reactive nitrogen emissions that may cause environmental problems such as $O_3$ pollution. To address this issue, we recommend applying an appropriate FAR that can optimize crop yields while minimizing reactive nitrogen emissions. In this study, field flux measurements reveal that the DF method can reduce soil HONO emissions compared to the SF method, which could be considered for future environmental policies. Moreover, nitrification inhibitors that are used to slow the nitrification process ($NH_4^+ \rightarrow NO_2^-$ $\rightarrow NO_3^-$) may also be able to reduce HONO emissions by reducing soil $NO_2^-$ production. However, this needs more laboratory experiments to quantify the emission reduction efficiency as well as side effects. In the future, comparative field experiments, as well as laboratory experiments that utilize different fertilization methods, different FAR, etc., should be conducted simultaneously to provide additional references for policymakers.

## 4 Uncertainty and future needs

Flux measurements with chamber methods have been widely used in determining the fluxes of greenhouse gases, $NO_x$, and some volatile organic compounds. HONO is more reactive than the above gases and can be produced during measurement. In terms of future OTDC development and application, improvements should align with the following aspects.

- Intercomparison with other methods. While several methods have been applied to measure soil HONO emissions, such as the REA, AG, and OTDC methods, intercomparisons between different methods have not been reported, which should be conducted to assess the reliability and accuracy of these methods.

- Simultaneous $NO_X$ measurements. $NO_X$ could be the most important species, which can lead to positive interference on the flux measurements. Its levels in the experimental and reference chambers could be different due to soil emissions. Simultaneous $NO_X$ (especially $NO_2$) measurements in the chambers allow an estimation of the potential interferences with the combination of uptake coefficients on soil and Teflon film surfaces.

- Correction of photolytic loss. Photolysis of high-concentration HONO in the experimental chamber can lead to HONO loss and then an underestimation of HONO fluxes. In addition to the residence time in the chamber, measurements of HONO photolysis frequency are required to correct the effects of photolysis.

- Translation of results obtained in this study to other sites. Here, we develop a relationship between HONO emissions and FAR for the first time. This relationship can be further used in estimating EF values and total reactive nitrogen emissions on a regional scale. Due to the limited field flux measurements, it can also provide preliminary insights into HONO emissions on a larger scale. Nevertheless, one should bear in mind that this relationship may vary with soil type, fertilizer type, climate conditions, plant impact, etc., which leads to uncertainties in the estimate in this study and needs future studies to verify.

**5 Conclusion**

This study presents measurements of HONO fluxes above agricultural fields in the North China Plain. Experiments are conducted simultaneously under two scenarios: normal fertilizer use (same as local farmers) and no fertilizer use, with three duplicated experiments conducted in parallel for each. The influencing factors and atmospheric implications of soil HONO emission were also discussed based on flux measurements in this and previous studies. The main conclusions are summarized as follows:

1) $F_{HONO-NP}$ and $F_{HONO-CK}$ show similar levels before fertilization, with negative values during nighttime, indicating that soil may occasionally act as a HONO sink at night. $F_{HONO-NP}$ can be largely enhanced

by nitrogen fertilizer use as its average increases to $97.7 \pm 8.6$ ng N m$^{-2}$ s$^{-1}$ after fertilization. However, as for other conditions (non-fertilized or long times after fertilization), the averaged HONO fluxes are in excellent agreement with most flux studies with small fertilizer application rates. Besides, the observed $F_{HONO-NP}$ always shows a bell-shaped diurnal variation, which is positively correlated to air temperature but opposite to relative humidity, implying their potential impacts on soil HONO emissions. Moreover, we find that HONO fluxes decline by ~30% after rainfall due to significant increases in soil water content.

2) Soil is an important and even dominant source of daytime HONO during short periods after intensive fertilizer application events (higher FARs). The observed HONO flux after fertilization can explain daytime HONO missing sources previously reported at this site and in other rural regions. Therefore, the synchronous measurements of fluxes and ambient concentrations are crucial to understanding the HONO budget as well as the follow-up atmospheric impacts on air quality, e.g., the regional abundance of $O_3$ and aerosol. Moreover, we found that deep-burying fertilizer can reduce soil HONO emissions compared to traditional spreading on the soil surface, constituting a potential HONO emission reduction strategy.

3) Thanks to the HONO fluxes measurement covering the whole crop growing season, for the first time, we estimated a HONO emission factor of $0.68 \pm 0.07\%$ related to the applied nitrogen. The emission factor is even comparable to that of NO and $N_2O$, suggesting a non-negligible role of nitrogen loss through HONO emission. Accordingly, the fertilizer-induced cumulative HONO emissions are estimated to be 22.3 and 60.8 Gg N yr$^{-1}$ from agriculture fields in the NCP and China mainland, respectively. Considering that nitrogen fertilizers are commonly used for agricultural fields and vegetable growing areas, this study also demonstrates the need for subsequent measurements of HONO fluxes on various underlying surfaces to accurately estimate reactive nitrogen (HONO, NO$_X$, etc.) emissions from those ecosystems.

In all, we demonstrate that soil HONO emissions and the promotion effect of fertilizer use should be considered in regional chemistry-transport models. It helps improve the air quality prediction and advance the understanding of the reactive nitrogen budget. It may also benefit future environmental policies in terms of mitigating regional air pollution.

**Data availability.** All the data used in this study are available at https://doi.org/10.5281/zenodo.8420431 (Song et al., 2023b) and upon request from the corresponding authors.

**Author contributions.** YZ and YM designed the experiments. YS carried out the experiments. YS and CX led the data analysis and manuscript writing with inputs from all co-authors. CX, YZ, PL, FB, XL, and YM revised the manuscript.

**Competing interests.** The contact authors declare that neither they nor their co-authors have any competing interests.

**Acknowledgments.** The authors thank Liwei Guan for his help during the field experiments.

**Financial support.** This work was financially supported by the National Natural Science Foundation of China (Grant No. 41931287, 42130714, and 41975164) and National Key Research and Development Program (Grant No. 2022YFC3701103). C.X. is thankful for the support of the Alexander von Humboldt Foundation.

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

**Table**

Table 1. Summary of the maximum values of HONO flux in field measurements over different soil types and corresponding measurement methods and fertilizer application rates (FAR) worldwide.

| Soil type | Method | HONO flux (ng N m$^{-2}$ s$^{-1}$) | | | FAR (kg N ha$^{-1}$) | References |
| --- | --- | --- | --- | --- | --- | --- |
| | | Mean | Max-1[a] | Max-2[b] | | |
| Agriculture | REA[c] | - | 7.0 | 1.4 | 0 | 1 |
| Forest | REA | 1.4 | 18.3 | 2.7 | 0 | 2 |
| Grassland | REA | - | 2.3 | 1.0 | 0 | 3 |
| Forest | AG[d] | 0.56 | 0.98 | - | 0 | 4 |
| Maize | AG | - | - | 2.3 | 33.4 | 5 |
| Wheat | AG | 0.84 | 15.4 | 2.8 | 69 | 6 |
| Maize | OTDC[e] | - | 1515 | - | 330 | 7 |
| Maize | OTDC | 21 | 40 | - | 45 | 7 |
| Maize | OTDC | - | 40 | 20 | 180 | 8 |
| Wheat | OTDC | 2.9 | 7.7 | 5.7 | 69 | 9 |
| Agriculture | OTDC | 34 | 348 | 83 | 247 | 10 |
| Maize | OTDC | 63 | 372 | 126 | 300 | This study |

[a]: maximum values in the time series; [b]: maximum values in the diurnal variations; [c]: relaxed eddy accumulation; [d]: aerodynamic gradient; [e]: open-top dynamic chamber.

1: (Ren et al., 2011); 2: (Zhou et al., 2011); 3: (von der Heyden et al., 2022); 4: (Sörgel et al., 2015); 5: (Laufs et al., 2017); 6: (Meng et al., 2022); 7: (Xue et al., 2019a); 8: (Tang et al., 2019); 9: (Tang et al., 2020); 10: (Xue et al., 2022b).

**Figures**

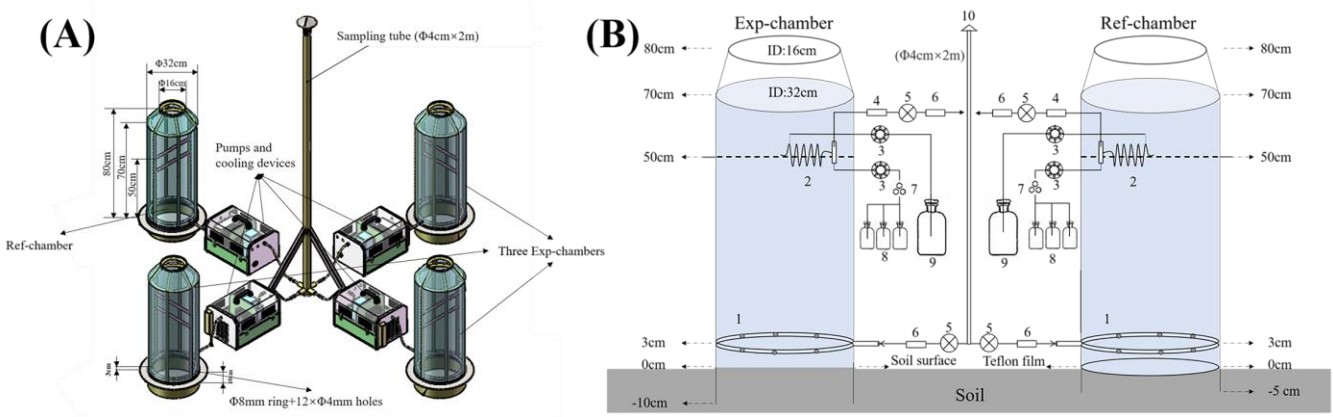

**Figure 1. The layout (A) and constructions (B) of the OTDC system and other equipment (upgraded based on Xue et al. (2019a)). 1. Stainless collar. 2. Stripping coil. 3. Peristaltic pump. 4. Drying tube. 5. Diaphragm pump. 6. Flow regulator. 7. Valve of 24 accesses. 8. Sample bottles. 9. Absorption solution bottle. 10. Sampling tube. Exp-chamber: experimental chamber; Ref-chamber: reference chamber.**

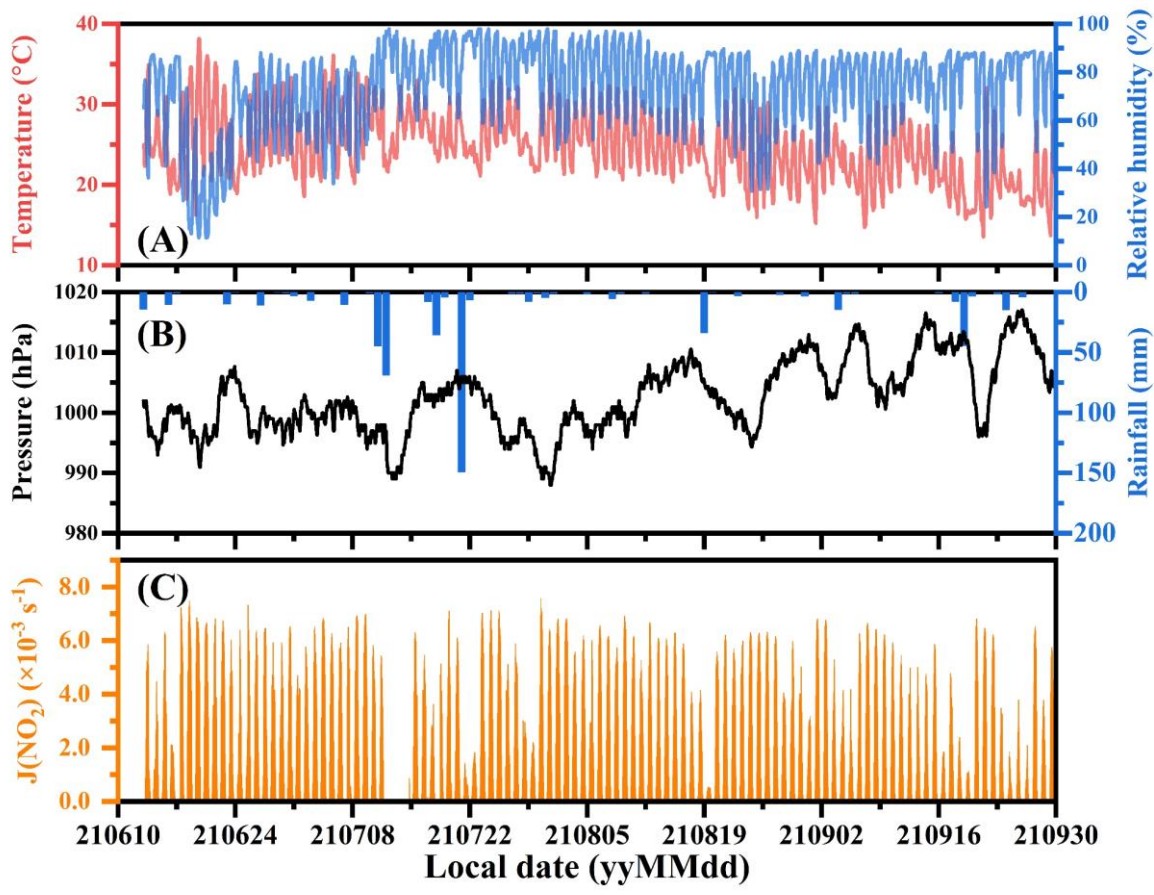

780

**Figure 2. The time series of meteorology (A: air temperature and relative humidity; B: air pressure and rainfall; C: the photolysis frequency of $NO_2$ ($J(NO_2)$)) during maize season at the experiment site.**

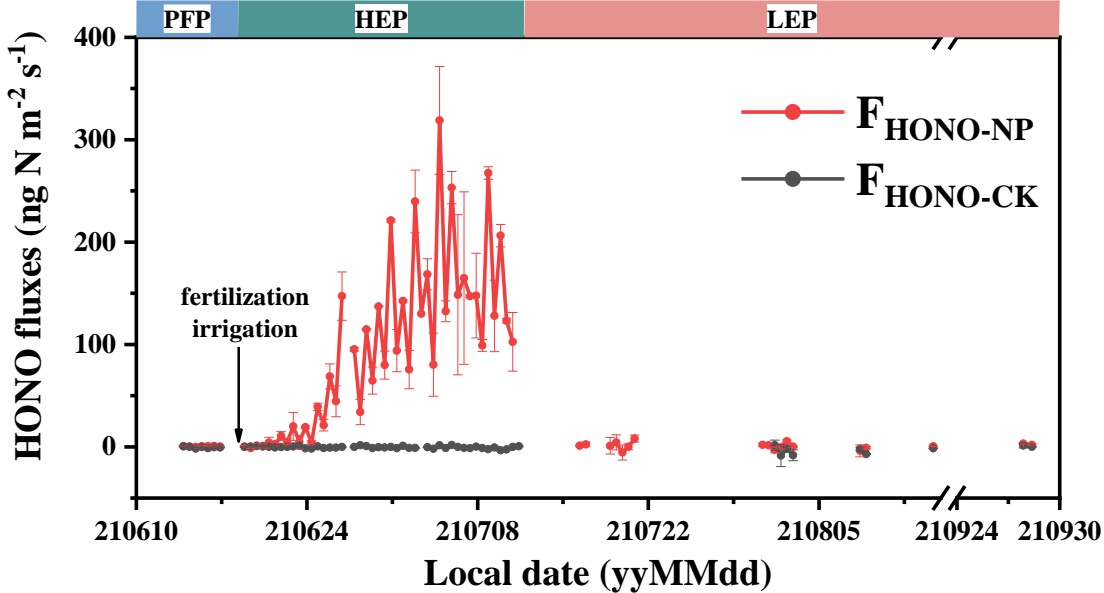

**Figure 3. Variations of HONO fluxes at 12 h intervals from NP and CK plots (F$_{HONO-NP}$ and F$_{HONO-CK}$) during the maize season 2021. CK: control, normal flood irrigation but no fertilization for decades; NP: fertilizer deep placement and normal flood irrigation, same as local farmers; PFP: pre-fertilization period, before June 18; HEP: high HONO emission period, from June 18 to July 10; LEP: low HONO emission period, after July 10. The error bars represent the standard deviations of HONO fluxes from NP or CK plots (n=3).**

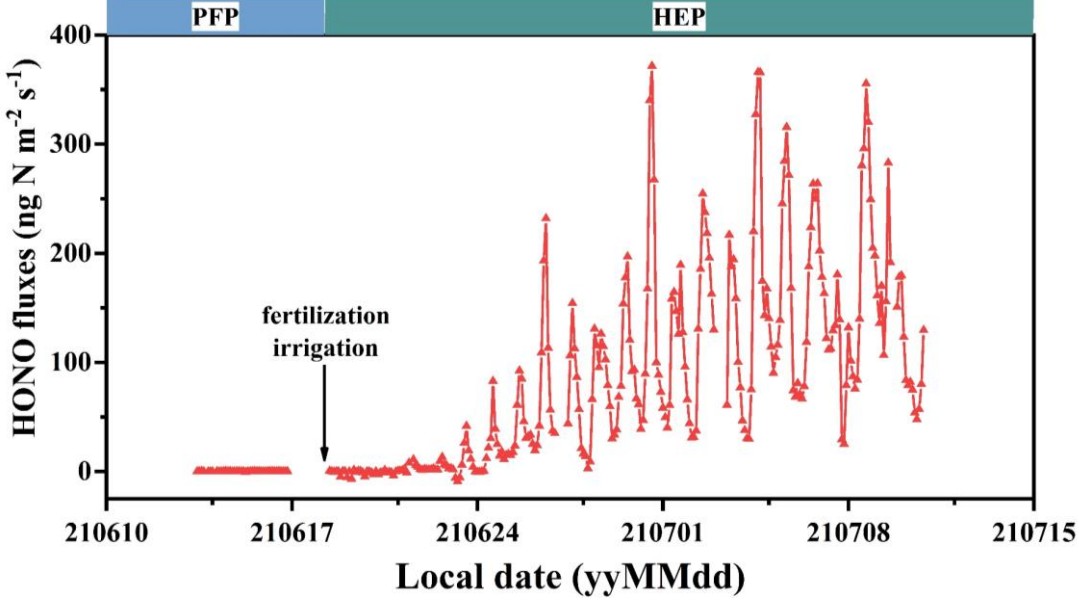

**Figure 4. Variations of 2-h interval HONO fluxes from the NP plots (F$_{HONO-NP}$) from June 13 to July 10, 2021. NP: fertilizer deep placement and normal flood irrigation, same as local farmers; PFP: pre-fertilization period; HEP: high HONO emission period.**

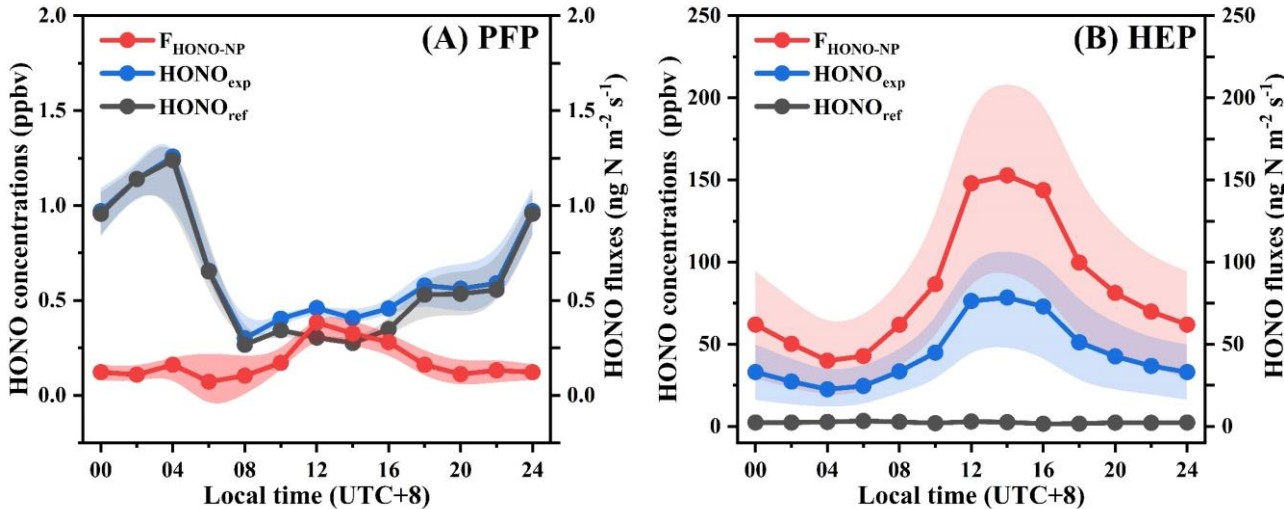

**Figure 5. Diurnal variations of HONO concentrations in Exp-chamber and Ref-chamber and HONO fluxes from the NP plot (HONO$_{exp}$, HONO$_{ref}$, and F$_{HONO-NP}$) during PFP and HEP. Shadows represent half of the standard deviation (±0.5 σ). NP: fertilizer deep placement and normal flood irrigation, same as local farmers; PFP: pre-fertilization period; HEP: high HONO emission period.**

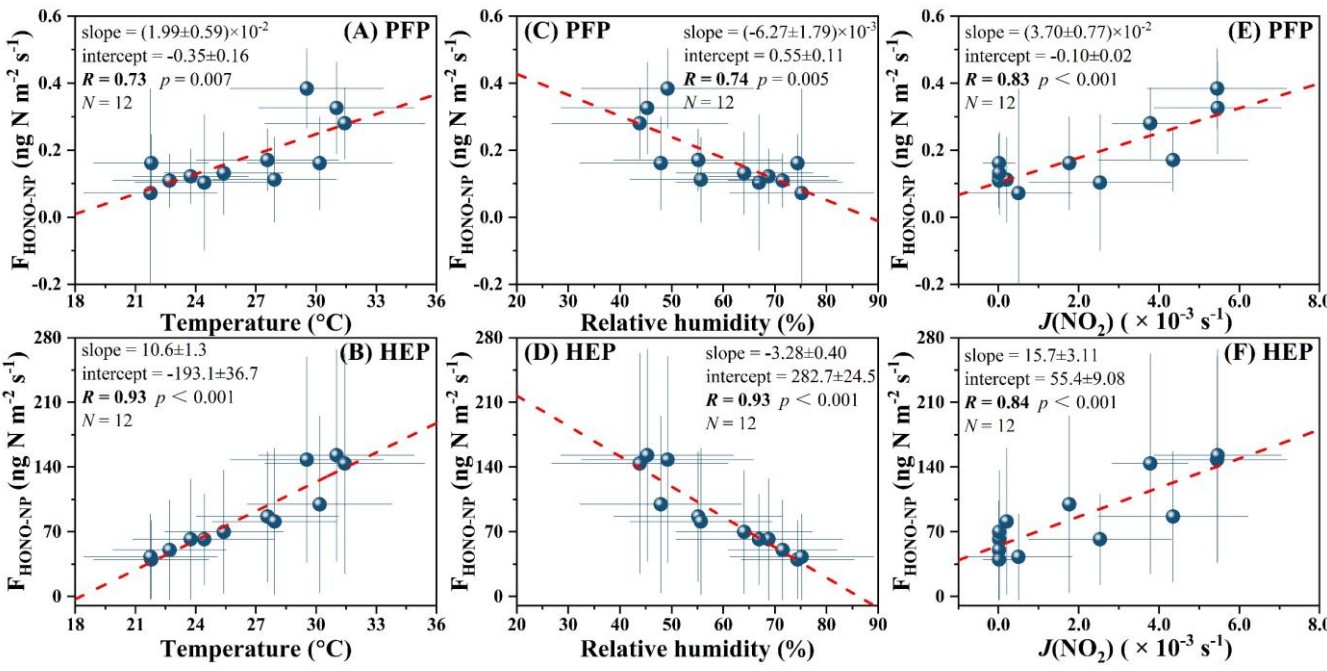

**Figure 6. Correlation of the diurnal F$_{HONO-NP}$ with the meteorological parameters (air temperature, air relative humidity, ad J(NO$_2$)) during PFP and HEP. F$_{HONO-NP}$: HONO fluxes from the NP plots; NP: fertilizer deep placement and normal flood irrigation, same as local farmers; PFP: pre-fertilization period; HEP: high HONO emission period.**

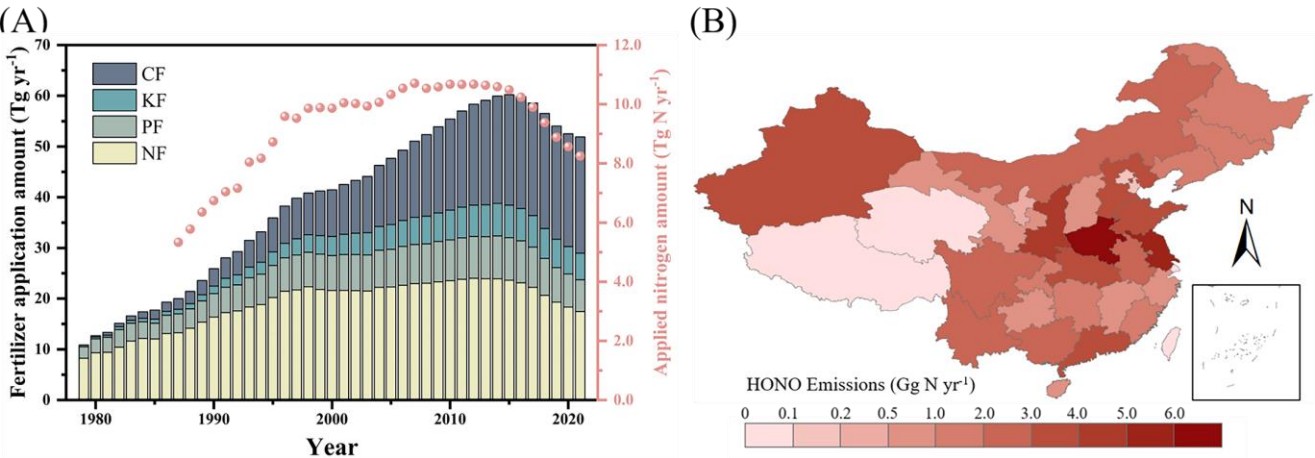

**Figure 7. (A): the national fertilizer application amount (CF: compound fertilizer; KF: potash fertilizer; PF: phosphatic fertilizer; NF: nitrogen fertilizer) in China during 1978–2021 and applied nitrogen amount (NF amount + 0.22 × CF amount) in the North China Plain during 1987–2021; (B): the annual HONO emissions from fertilized fields in 2021 in China (HONO emission factors changed exponentially with fertilizer application rate in each province). Data source: China Statistical Yearbooks 1979–2022.**

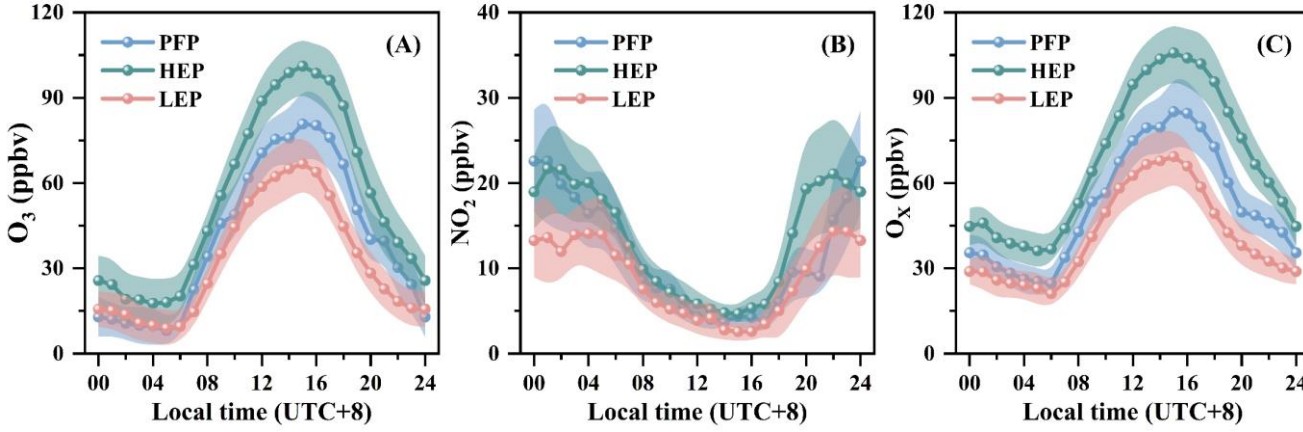

**Figure 8. The diurnal variations of $O_3$, $NO_2$, and $O_X$ ($O_3 + NO_2$) concentrations during PFP, HEP, and LFP. Shadows represent half of the standard deviation (±0.5 σ). PFP: pre-fertilization period; HEP: high HONO emission period; LEP: low HONO emission period.**
