# Peer review of "Measurement report: Exchange fluxes of HONO over agricultural fields in the North China Plain"

_EGUsphere, 2023_

## Author Comment (AC1)

Dear Editor, dear reviewers,

5    Many thanks for your valuable comments and thoughtful suggestions that help to improve the quality of this manuscript. The following are our responses to your comments.

The comments are shown in black, our responses to the comments are presented in blue, and the new or modified texts are provided in red.

**Response to Reviewer #1**

This paper studied soil HONO emissions from fertilized soil in Chinese agricultural land. In general, the authors provide some interesting field flux data, and discussed potential influencing factors and atmospheric implications.

15    I have some comments need to be addressed before it can be accepted.

**Response:** Thank you for your positive evaluation of our work. The following are our responses to your comments.

**Comment 1:**

20    Introduction, I would suggest the authors focus on literature review in fertilization caused soil HONO emissions, both in field and laboratory studies. The review in methods of HONO flux measurement are not necessary.

**Response:** Thanks for the suggestion. It is indeed necessary to briefly summarize current soil HONO emission studies. So, in addition to Table 1 where field HONO flux measurements worldwide are shown,

25    we shortened the texts about HONO flux measurement methods and added new sentences as a brief literature review of soil HONO emissions.

L75–L106 in the change-tracked version:

"Flux measurement can provide direct evidence about the production and/or deposition of HONO on the ground surface (von der Heyden et al., 2022; Xue et al., 2022). The aerodynamic gradient (AG) and

relaxed eddy accumulation (REA) methods have been developed and applied to HONO flux measurements in recent years, providing a good option to measure HONO flux (Laufs et al., 2017; Sörgel et al., 2015; von der Heyden et al., 2022; Zhou et al., 2011). As shown in Table 1, Laufs et al. (2017) and Sörgel et al. (2015) measured HONO fluxes using the AG method above bare soil, different crops, and forests, and they found the flux was mainly contributed by $NO_2$-related photosensitized reactions. Ren et al. (2011), Zhou et al. (2011), and von der Heyden et al. (2022) conducted flux measurements using the REA method above agricultural fields, forests, and grassland, respectively, and the maximum HONO fluxes in these studies were all less than 20 ng N $m^{-2}$ $s^{-1}$. Apart from above meteorological methods, the dynamic chamber method uses ambient air to flush the chamber to avoid the formation of water film, allowing the determination of HONO exchange fluxes between soil and atmosphere (Tang et al., 2019; Xue et al., 2019). Different from the above measurements, Xue et al. (2019) and Xue et al. (2022) observed extremely high levels of HONO fluxes under "over-fertilized" conditions, with the maximum fluxes being 1515 and 348 ng N $m^{-2}$ $s^{-1}$, respectively, which approach even exceed the high values in the laboratory experiments (Oswald et al., 2013; Wang et al., 2021). Besides, the mechanism of soil HONO emissions is still incomplete understood, e.g., the dominant source of soil nitrite (Bhattarai et al., 2021; Ermel et al., 2018; Oswald et al., 2013; Scharko et al., 2015; Song et al., 2023; Wu et al., 2019), the chemical-physical transformation of soil nitrite to gas-phase HONO (Bao et al., 2022; Kim and Or, 2019; Oswald et al., 2013; Su et al., 2008; Xue et al., 2022). Moreover, available flux measurements are still limited and most of them were conducted over a short period of normally less than one month. A systematic and relatively longer measurement covering a whole growing season of a crop is lacking, resulting in limitations in estimating the HONO emission factor ($EF_{HONO}$) as well as the understanding of the reactive nitrogen budget at an annual scale (Xue et al., 2022).

**Comment 2:**

L173, the unit should be $m^2$.

**Response:** This typo has been corrected in the revised manuscript.

**Comment 3:**

L177, the calculation method of cumulative HONO-N emissions should be introduced. Many data points were missed in Figure 3, how did you fill these gaps?

**Response:** We added new sentences in the revised manuscript to introduce how we fill the gaps when calculating cumulative HONO emissions.

L397–L404 in the change-tracked version:

"During HEP, as shown in Fig. 3, the flux not only shows a distinct diurnal variation but also a varying daily peak relative to days after fertilization. Therefore, we conducted intensive flux measurements with a 2-h resolution to capture the diurnal variation and establish the average flux. The latter is subsequently used in calculating the cumulative HONO emissions during HEP. And the missing data during this period were filled in by time interpolation. In contrast, fluxes during LEP show no significant daily variation. Hence, the average of measurements with 12-h resolution can effectively characterize the emission level during LEP, and the resulting average flux is further used in calculating the cumulative HONO emissions during this period."

**Comment 4:**

Soil temperature data are encouraged to provide during field HONO measurements.

**Response:** As suggested, measurements of soil temperature are provided in Fig S2:

[Figure]

Figure S2. Variations of soil water-filled pore space (WFPS) and rainfall (A), surface soil (5 cm) temperature (B), and concentrations of soil $NH_4^+$-N and $NO_3^-$-N from the NP (chemical N fertilizer and normal irrigation), and CK (no

fertilization but with normal irrigation) plots (C) during the maize growing season. Error bars represent the standard deviations (n=5 for soil WPFS and 3 for soil $NH_4^+$-N and $NO_3^-$-N concentrations).

80

**Comment 5:**

L257-260, these sentences can be moved to the Methods section.

**Response:** Corrected as suggested.

85 **Comment 6:**

L323-324, another reason could be high soil moisture decrease gas diffusion in soil profile.

**Response:** We added a new sentence regarding this point.

L342–L345 in the change-tracked version:

"Besides, high soil moisture after rainfalls will decrease gas diffusion in the soil profile, which could not

90 only mitigate HONO transportation in the soil but also inhibit soil nitrification, the most important microbial process for HONO production (Scharko et al., 2015), by decreasing $O_2$ supplement, thus reducing HONO emissions (Wu et al., 2019)."

**Comment 7:**

95 L380, delete the word of decreasing.

**Response:** Corrected as suggested.

**Comment 8:**

L382-383, we estimate a relationship between the emission factor caused by fertilization and fertilization

100 amount, please see the reference Wu et al., 2022 JGR: Atmospheres, 127, e2021JD036379. I would suggest the authors can discuss the results in this manuscript with our results.

**Response:** Our estimate is much lower than that of Wu et al. We explained in detail about the discrepancy in the revised manuscript.

L434–L449 in the change-tracked manuscript:

105    Compared to a recent study (Wu et al., 2022), in which HONO emission of 0.52 Tg N yr$^{-1}$ on a national scale is estimated, our estimate is much lower. The large discrepancy could be due to the differences between those two studies.

1) Soil area difference: this study only considers the agricultural soils with nitrogen fertilizer application, $1.7 \times 10^6$ km$^2$ (data source: the China Statistical Yearbooks 2022,

110     http://www.stats.gov.cn/sj/ndsj/2022/indexch.htm, last access: October 6, 2023), which is only one-sixth of the national land area.

2) Time scale difference: this study aims to quantify the fertilizer effect and the significant positive fluxes are only observed within ~1 month after fertilizer application, while Wu et al. (2022) estimate annual emissions.

115   3) Bi-directional effect: soil-atmosphere is a bi-directional process, e.g., the soil surface can release or absorb HONO (Bao et al., 2022; VandenBoer et al., 2013), which can also be verified by the occurrence of positive and negative fluxes in this and other studies (Table 1). Current flux measurements measured the net flux of the bi-directional soil-atmosphere exchange, while the calculated estimate only considers the soil-to-atmosphere process (Weber et al., 2015; Wu et al.,

120     2022).

**Comment 9:**

L445, I would not call it long-period measurements if the authors only showed ~ 1 month available data.

**Response:** "long-period" has been removed.

125

**Comment 10:**

Figure 5 is confusing with two units. The authors should separate them with different axis.

**Response:** Improved as suggested.

[Figure]

**Figure 5.** Diurnal variations of HONO concentrations in Exp-chamber and Ref-chamber and HONO fluxes from the NP plot (HONO$_{exp}$, HONO$_{ref}$, and F$_{HONO-NP}$) during PFP and HEP.

**Comment 11:**

The authors cited too many of their own works, while lacking compare with other studies, such as the work from groups of Jonathan Raff, Marja Maljanen, and Dianming Wu. I would suggest the authors can discuss and compare their results with that from other scientists.

**Response:** Thanks for the comments. We added and discussed more studies from the above-mentioned groups in the revised manuscript. For example, two works from Dianming Wu (Weber et al., 2015; Wu et al., 2022) were added to the regional estimation section, which was detailed discussed in R1C8; three studies from Jonathan Raff (Scharko et al., 2015), Marja Maljanen (Bhattarai et al., 2021) and Dianming Wu (Song et al., 2023) were added in the Introduction section and Section 3.5.1 to highlight the importance of nitrification and denitrification in soil HONO production. Another work from Marja Maljanen (Maljanen et al., 2016) was added in Section 3.6.2 to compare the HONO emissions from soils to those from livestock dung.

**Response to Reviewer #2**

This study by Song et al. reports long-period measurements of HONO fluxes above agricultural fields in the North China Plain (NCP). Experiments are conducted on both normal fertilizer use and no fertilizer use fields. The reported soil HONO emissions are carefully compared with existing literature. The influencing factors and atmospheric implications (nitrogen budget, ozone air quality) of soil HONO emission are also analyzed.

This study fills the largely missing measurement of soil HONO emissions during fertilizer periods in the NCP, which is much underappreciated in current studies of air quality in this region. This is a valuable and significant contribution to the community. The analyses are comprehensive and informative. This is also a well-structured, well-detailed, and well-written manuscript. I recommend publication in ACP after some minor revisions.

**Answer:** Thank you for your positive evaluation of our work. The following are our responses to your comments.

**Comment 1:**

My little concern is about the application of the measured soil HONO emission factor (0.68%) to estimate the national scale soil HONO emissions (Section 3.6.2). While this estimate is important and valuable, I wonder to what extent this emissions factor can be applied to other regions? As the fertilizer type, approach, and meteorological parameters differ in other regions with NCP, the emission factor would likely not be constant, then the national estimate might be biased. Some discussions of the limitation of this approach and comparison with other studies on the national scale might be helpful.

**Response:** Yes, we do agree that the national estimate has uncertainties due to various factors, such as those mentioned by the reviewer. First, we updated the estimation method according to reviewers' comments (see below details); second, we added a new section, "4 Uncertainty and future needs"

Updated estimation method at L410–L459 in the change-tracked manuscript:

[revised manuscript text omitted]

**Comment 2:**

Section 3.6.3: does it also include ozone contributed by soil NOx (not solely from soil HONO)? Please clarify.

**Response:** Yes, the elevation of ozone concentrations may also result from soil $NO_X$ emissions in addition to HONO emissions. We clarified this in the revised manuscript as "Fertilized soils could also emit $NO_X$, which may also aggravate $O_3$ pollution." Besides, we added a recent reference that highlights the influence of soil HONO and $NO_X$ emissions on regional $O_3$ concentrations as "A recent estimation revealed that daily maximum 8 h average $O_3$ enhancement from soil HONO and $NO_X$ emissions of 8.0 ppbv over the NCP and 5.5 ppbv over China in June–July 2019 (Tan et al., 2023)."

**Comment 3:**

Line 254: "it is worth noting that the high water content but no fertilization for the CK plots may contribute to the negative fluxes." It is not clear to me (and possibly other readers) how high water content would lead to negative flux. Please clarify.

**Response:** Reasons may lead to the negative fluxes for the CK plots after irrigation: 1. Unfertilized soils produce less HONO than fertilized soils; 2. After irrigation, the soil is relatively moist. Higher soil moisture will accelerate the diffusion of soil mineral nitrogen, dilute the concentration of $NO_2^-$ in the soil, and reduce the concentration of substrate for acid-base reaction between $NO_2^-$ and $H^+$; 3. Higher soil moisture is also unfavorable to the conversion from liquid phase $HNO_2$ to gas phase HONO in soil; 4. Moist soils, especially topsoil, make it easier for the atmospheric HONO to deposit to the land. Combined with the above points, unfertilized soil after irrigation is more likely to exhibit negative fluxes. This sentence was rewritten as "Nevertheless, it is worth noting that the high soil moisture after irrigation for the unfertilized CK plots may reduce the conversion from liquid phase $HNO_2$ to gas phase HONO in soils and accelerate the absorption of atmospheric HONO at the soil surface, thereby contributing to the negative HONO fluxes."

**Response to Reviewer #3:**

In the manuscript by Song et al. HONO fluxes from freshly fertilized soil surfaces are compared to non-fertilized soils in open-top dynamic chambers (OTDC) in China. Very high fluxes were determined for fertilized soils in good agreement with previous similar experiments, which were however much higher compared to all other HONO flux measurements by the gradient and REA methods under normal fertilization conditions (see below). Since fluxes give more direct information on ground surface sources and sinks compared to former PSS approaches, HONO flux measurements are highly recommended. The manuscript contains interesting results (e.g. for the first time cumulative HONO emissions after fertilization) and also valuable recommendation how to reduce HONO emissions by fertilizer usage and should be published after my concerns have been considered.

**Response:** Thanks for your great efforts and valuable comments below, which helped to improve our manuscript. The following are our point-to-point responses to your comments.

**Major concerns:**

1) Method used

In Table 1 different flux results are compared, nicely showing that in most OTDC studies significantly higher fluxes were determined compared to well-established open flux measurements like the aerodynamic gradient and relaxed Eddy accumulation methods. There are two possible explanations for this observation:

a) Fluxes in open dynamic chambers are overestimated by different potential artifacts. First, the temperature in a Teflon chamber and of the soil surface – even in open dynamic chambers – will be significantly higher compared to real open surfaces by the greenhouse effect (ca. 4 min residence time, ca. 1000 W/m2 at noon, low heat capacity of the air…). This will not only increase biological activity, but has also an influence on the soil surface coverage by water affecting surface exchange processes. Higher temperature decreases the Henry's law constant of HONO and reduces water adsorption, decreasing the volume of water for solution. Both will increase HONO fluxes. This cannot be considered for by using a reference chamber and/or by comparing fertilized with non-fertilized soils as done in the present study (which I highly appreciate!). Second, by changing surface humidity and temperature also artificial heterogeneous HONO formation on chamber surfaces may be enhanced (ca. 4 min residence

time). Typically, this is aimed to be considered for by using the difference to the reference chamber covered with a Teflon foil on the ground. However, if for example soils emit electron-rich VOCs and NOx (for the latter see the results of the present study), which are known to heterogeneously form HONO (see e.g. George et al., 2005) than the artificial heterogeneous HONO formation on chamber surfaces will be higher in the soil chamber compared to the reference chamber. And fertilized soils will emit more NOx compared to non-fertilized soils. Also here the comparison of fertilized with non-fertilized soils may not help, since e.g. NOx emissions may be much higher on fertilized soils. Furthermore, irradiated Teflon surfaces are known to artificially form HONO under irradiation, which is still not completely understood (see e.g. Rohrer et al. 2005; doi: 10.5194/acp-5-2189-2005). The latter artifact however, may be considered for in the present study by the use of the reference chamber.

In conclusion to a) larger surfaces/chambers always bear the risk of overestimating HONO fluxes! Thus, I would highly recommend an intercomparison of the OTDC method with direct AG or REA measurements in the future (before publishing more OTDC results…) and at least to highlight in the present study that the fluxes determined in OTDCs may be overestimated.

**Response:** In fact, we are constantly making improvements to the OTDC system. Compared with the original OTDC in Xue et al. (2019), we added a sampling tube in each OTDC system to ensure that the flushing gas in four chambers is consistent, which all come from the atmosphere at 2 m, which is less affected by soil emissions. Besides, we also added a cooling device for each diaphragm pump to keep the temperature of flush gas close to the air temperature.

As far as we know, the REA method has not been applied in measuring HONO flux in China, and the AG method has only been used once (Meng et al., 2022). This indicates that the meteorological methods are not mature for HONO flux measurement.

In our recent study, the HONO flux in the laboratory and the field are similar under the same fertilizer application amount (Xue et al., 2022). In addition, there is also a lot of evidence from field observations to prove the high HONO emission after fertilization (Liu et al., 2019; Xue et al., 2021). All these indicate that there are non-negligible HONO emissions from fertilized soils. In general, the effects of fertilization are often less than a month, which can be confirmed by many flux observations (Tang et al., 2019; Tang et al., 2020; Xue et al., 2022). Soil emissions are not a major source of HONO during the non-fertilization period or a long time after fertilization. For example, negative fluxes were measured many times before

fertilization in this study, and the contribution of soil emissions in winter HONO budget analysis was negligible in previous studies (Xue et al., 2020; Zhang et al., 2022).

320   Nevertheless, we admit that the OTDC method is still associated with uncertainties and should be further improved. We then add a new section to highlight the uncertainties.

L496–L520 in the change-tracked manuscript:

**4 Uncertainty and future needs**

Flux measurements with chamber methods have been widely used in determining the fluxes of greenhouse
325   gases, $NO_x$, and some volatile organic compounds. HONO is more reactive than the above gases and can be produced during measurement. In terms of future OTDC development and application, improvements should align with the following aspects.

- Intercomparison with other methods. While several methods have been applied to measure soil HONO emissions, such as the REA, AG, and OTDC methods, intercomparisons between different
330     methods have not been reported, which should be conducted to assess the reliability and accuracy of these methods.

- Simultaneous $NO_X$ measurements. $NO_X$ could be the most important species, which can lead to positive interference on the flux measurements. Its levels in the experimental and reference chambers could be different due to soil emissions. Simultaneous $NO_X$ (especially $NO_2$)
335     measurements in the chambers allow an estimation of the potential interferences with the combination of uptake coefficients on soil and Teflon film surfaces.

- Correction of photolytic loss. Photolysis of high-concentration HONO in the experimental chamber can lead to HONO loss and then an underestimation of HONO fluxes. In addition to the residence time in the chamber, measurements of HONO photolysis frequency are required to
340     correct the effects of photolysis.

- Translation of results obtained in this study to other sites. Here, we develop a relationship between HONO emissions and FAR for the first time. This relationship can be further used in estimating EF values and total reactive nitrogen emissions on a regional scale. Due to the limited field flux measurements, it can also provide preliminary insights into HONO emissions on a larger scale.
345     Nevertheless, one should bear in mind that this relationship may vary with soil type, fertilizer type,

climate conditions, plant impact, etc., which leads to uncertainties in the estimate in this study and needs future studies to verify.

b) The higher HONO fluxes may be caused by the very high fertilizer application amounts of ca. 300 kg N / ha applied (see line 142), which is much higher compared to typical amounts of <100 kg N/ha applied in most regions of the world and on soils in other HONO flux studies (see table 1). This is especially important, since the authors observed an extremely nonlinear relationship between the HONO fluxes and the nitrogen application amount (see the nice results shown in Figure S6 in Xue et al., 2022a, which will bring the very different published flux results together). From this exponential relationship HONO fluxes at 300 kg N/ha will be almost two orders of magnitude higher compared to fluxes at the average global application amount of 75 kg N/ha (see reference in Xue et al., 2022a), which is exactly the ratio observed between former flux studies and those in the recent OTDC studies (see Table 1). This nonlinear relationship should be highlighted and discussed to explain the different flux results. In addition, the authors should consider the exponential relationship for estimation of regional and global HONO emissions from fertilized soils. Here much lower contribution will be obtained for any typical fertilization amount.

**Response:** Our study supports that the main reason for the huge difference in HONO flux between several OTDC studies in our site and others is the difference in fertilization amount. Our previous study showed an exponential relationship between the fertilization amount and HONO flux (Xue et al., 2022). Therefore, we emphasize this nonlinear relationship in Section 3.4 and use it to explain the apparent difference in flux from previous studies as "With the increase of fertilizer application rates (FAR), the emission of nitrogenous gas tends to show nonlinear growth (Grant et al., 2006; Ma et al., 2010; McSwiney and Robertson, 2005; Shcherbak et al., 2014; Xue et al., 2022). Except for previous measurements at this agricultural site (Tang et al., 2019; Xue et al., 2022; Xue et al., 2019), fluxes from all other measurements are much lower than in this study, which should be due to the difference in FAR."

From a global perspective, the FAR varies largely with regions, indicating the potential extended application of the relationship between FAR and $F_{HONO\_max}$. FAR in China is relatively higher, but

considerable regions also have similar FAR levels (Figure R1), suggesting significant HONO emissions in these regions.

Regarding the estimate of EF and total emissions on a regional/national scale, please see the detailed response to the following comment.

[Figure]

Figure R1. A summary of worldwide nitrogen fertilizer use per hectare of cropland in 2019 (Source: Our World in Data and UN Food and Agricultural Organization (FAO)).

2) Estimated implication:

This brings me to my second major concern, the estimated HONO emissions for China. In their implication section 3.6 the authors used a constant EF(HONO) of 0.68% HONO emitted per amount of N-fertilizer applied. However, if the HONO fluxes are exponentially increasing with the fertilizer amount (see again Figure S6 in Xue et al., 2022a) a constant ratio cannot be used! Since this ratio was determined at very high fertilizer amount of 300 kg N /ha the average EF(HONO) will be much lower, considering variable application rates typically lower than that extreme value, even in China (see e.g. Figure 2a in Potter et al., 2010, DOI: 10.1175/2010EI288.1). The authors should apply the exponential relation considering regional variable fertilizer application amounts. In addition, the authors should estimate HONO fluxes from this relationship for typical global nitrogen application amounts (<100 kg N /ha) to highlight that the results are only applicable to the more extreme Chinese fertilization conditions. Here I expect at least one order of magnitude lower HONO emissions.

The results shown in Figure S6 in Xue et al., 2022a may also identify "over-fertilization", for which nitrite formation by biological ammonium oxidation may be faster than the biological nitrite oxidation. In this case nitrite may accumulate and HONO emissions may be controlled by solubility in soil water (see discussion of the temperature and humidity dependence in section 3.5.2) which should be avoided! Thus, the authors may use these results to also recommend more environment-friendly fertilization amounts in the future for China (similar to most other regions of the world) besides their recommendation of the DF method.

**Response:** Thank you for the nice suggestion of combining the FAR-$F_{max}$ relationship with EF.

In the revised manuscript, we took the exponential relationship into account as mentioned, and re-estimated HONO emissions in China mainland and the NCP region. It is worth noting that the relationship in Xue et al. (2022) was between FAR and $F_{max}$, which can be extended to FAR and cumulative emissions when assuming that $F_{max}$ is proportional to cumulative emissions. However, FAR needs to be included in calculating again when considering the relationship between EF and FAR (EF = cumulative emission / FAR). In this case, the EF obtained at lower FAR will be smaller than at higher FAR, but not by an order of magnitude smaller. Then we found that the EFs in each province in China mainland were in the range of 0.15–0.5%. Accordingly, the fertilizer-induced cumulative HONO emissions are estimated to be 22.3 and 60.8 Gg N yr$^{-1}$ from agriculture fields in the NCP and China mainland, respectively. The following is the detailed description of the new estimation method and results.

L410–L459 in the change-tracked manuscript:

On a national scale, although the total fertilizer application amount has shown a downward trend after 2015, the current fertilizer application amount is still higher than that in 1978 by a factor of 5 (Figure 7A). Assuming nitrogen accounts for 22% of the compound fertilizer (N: $P_2O_5$: $K_2O$ = 28%: 6%: 6%), the total applied nitrogen (TN) and FAR in each province in 2021 were listed in Table S2 (data source: the China Statistical Yearbooks 2022, http://www.stats.gov.cn/sj/ndsj/2022/indexch.htm, last access: September 21, 2023). The FARs in each province are in the range of 40–250 kg N ha$^{-1}$. The relationship between the $F_{max}$ under SF condition ($F_{max-SF}$) and FAR was fitted based on our previous study, in which flux measurements were conducted in different FAR conditions (Xue et al., 2022) (Figure S5, $F_{max-SF}$ = 29.54 × exp (FAR / 98.04 – 20.19, FAR range: 0-250 kg N ha$^{-1}$). The reduction proportion of DF to SF method was calculated by substituting the FAR in this study (300 kg N ha$^{-1}$) into the above formula and comparing

it with the measured maximum flux (372 ng N m$^{-2}$ s$^{-1}$). Accordingly, the F$_{max}$ under DF condition (F$_{max\text{-}DF}$) is ~61% of F$_{max\text{-}SF}$ under the same FAR, indicating that the DF method can reduce 39% HONO emissions compared to the SF method. Besides, we assume that the HONO cumulation emission is proportional to F$_{max}$. Therefore, the EF$_{HONO}$ (%) at different FAR levels can be calculated by:

$$F_{max\text{-}DF} = F_{max\text{-}SF} \times 0.61 \qquad\qquad (eq\text{–}3)$$

$$EF_{HONO} = 0.68 \times \frac{F_{max\text{-}DF}}{372} \times \frac{300}{FAR} \qquad\qquad (eq\text{–}4)$$

Note that 0.68 and 372 represent the aforementioned EF, measured F$_{max\_DF}$ at a FAR of 300 kg N ha$^{-1}$, respectively. Based on (eq-3) and (eq-4), the EF$_{HONO}$ is in the range of 0.15–0.5% at a FAR range of 40-250 kg N ha$^{-1}$. Figure 7B shows the regional fertilizer-induced soil HONO emissions in each province, and the national soil HONO emission is around 60.8 Gg N yr$^{-1}$. Compared to a recent study (Wu et al., 2022), in which HONO emission of 0.52 Tg N yr$^{-1}$ on a national scale is estimated, our estimate is much lower. The large discrepancy could be due to the differences between those two studies.

1) Soil area difference: this study only considers the agricultural soils with nitrogen fertilizer application, $1.7 \times 10^6$ km$^2$ (data source: the China Statistical Yearbooks 2022, http://www.stats.gov.cn/sj/ndsj/2022/indexch.htm, last access: October 6, 2023), which is only one-sixth of the national land area.

2) Time scale difference: this study aims to quantify the fertilizer effect and the significant positive fluxes are only observed within ~1 month after fertilizer application, while Wu et al. (2022) estimate annual emissions.

3) Bi-directional effect: soil-atmosphere is a bi-directional process, e.g., the soil surface can release or absorb HONO (Bao et al., 2022; VandenBoer et al., 2013), which can also be verified by the occurrence of positive and negative fluxes in this and other studies (Table 1). Current flux measurements measured the net flux of the bi-directional soil-atmosphere exchange, while the calculated estimate only considers the soil-to-atmosphere process (Weber et al., 2015; Wu et al., 2022).

Considering that the NCP region consumes a large portion of fertilizer in China and suffers severe O$_3$ pollution, soil HONO emissions need to be a particular concern in this region. As shown in Figure 7, the TN and the corresponding soil HONO emissions were 8.0 Tg N and 22.3 Gg N yr$^{-1}$ in the NCP in 2021, respectively. The soil HONO emission is more than 10% of the annual regional soil NO$_X$ emissions in the

NCP ($0.18 \pm 0.01$ Tg N yr$^{-1}$) (Lu et al., 2021), indicating that fertilized soils are important primary sources of both HONO and NO$_X$, exceeding other primary HONO emissions, such as livestock dung (Maljanen et al., 2016; Zhang et al., 2023), and affecting regional air quality in the NCP. It may also indicate that soil HONO emissions should be considered in environmental policies to mitigate regional air pollution.

[Figure]

Figure S5. Correlations of fertilization application rates (FAR) and maximum HONO flux under surface fertilization ($F_{max-SF}$).

[Figure]

Figure 7. (A): the national fertilizer application amount (CF: compound fertilizer; KF: potash fertilizer; PF: phosphatic fertilizer; NF: nitrogen fertilizer) in China during 1978–2021 and applied nitrogen amount (NF amount + 0.22 × CF amount) in the North China Plain during 1987–2021; (B): the annual HONO emissions from fertilized fields in 2021 in China (HONO emission factors changed exponentially with fertilizer application rate in each province). Data source: China Statistical Yearbooks 1979–2022.

It is very complicated to recommend a proper FAR. As for air quality, smaller FAR leads to fewer emissions and mitigates air pollution. However, FAR significantly affects crop yields, another important issue we should care about. A proper FAR should be environmentally friendly and also align with the needs of crop production. To address this, we need to assess the benefits or losses at different FAR levels and find a balance between them, to achieve sustainability development.

**Minor comments:**

Minor comments in the order how they appear in the manuscript:

Line 51: please first use a reference to Song et al., 2022a before Song et al., 2022b (and exchange both in the list…). The same for other references, e.g. line 54: Xue et al., 2022b is used before Xue et al., 2022a.

**Response:** Changed as suggested.

Line 53: Better use the reference Kleffmann, 2007 (doi: 10.1002/cphc.200700016), since in the referred 2003 study daytime sources were not a main focus.

**Response:** Changed as suggested.

Line 56 and 57: Better use here only studies in which the daytime source could be quantified based only on experimental PSS data and not on uncertain assumptions. Here the first studies in which the PSS and the unknow HONO source could be quantified fully by experimental data were by Kleffmann et al., 2005 (doi: 10.1029/2005GL022524) and Acker et al., 2006 (doi:10.1029/2005GL024643). Thus, I would delete Kleffmann et al., 2003, Sparataro et al., 2013 and Su et al., 2008 (no exp. OH data, OH source by HONO only estimated)

**Response:** This is a good suggestion. The cited references were changed as suggested.

Line 68: Add reference by Tang et al., 2020.

**Response:** Improved as suggested.

Line 78: use "von der Heyden et al." throughout the text.

**Response:** Revised as suggested.

Line 78: add references by Ren et al., 2011, Zhou et al., 2011, Laufs et al., 2017.

**Response:** Added as suggested.

Line 168, equation-1: First, the units are missing for the factor 1/60 (min/s) and second, a factor 109 ng/g

is missing. But I would recommend to delete all factors and simply give SI units for all terms (i.e. F in

m3/s; M in ng/mol, P in Pa, R in J/mol/K).

**Response:** The equation and units were changed as suggested.

Line 172: K-1

**Response:** Revised**.**

Line 186, Figure S1: The shown correlation is not "high" but may be fair (see scatter in Figure S1). In

addition, the intercept will be zero by definition ($J(NO2) = 0$ during night…). And finally, a measure for

the actinic flux ($J(NO2)$) will not linearly correlate with a measure of the irradiance ("solar radiation").

Here a nonlinear correlation is expected since only for the irradiance the cos-dependence has to be

considered (check with the TUV model). A quadratic dependence forced by zero will better describe the

relation between both terms.

**Response:** Thank you for your advice. The fitting between $J(NO_2)$ and solar radiation and analysis

involving $J(NO_2)$ have been modified accordingly. After modification, we find that the correlation

coefficient between these two terms is greater than 0.9, so we still use "high" to describe their correlation.

Figure S1 was changed as follows:

[Figure]

**Figure S1.** Correlation between the measured J(NO$_2$) and solar radiation (August 28–September 30, 2021).

520 Line 193: Isn't the soil particle density depending on the soil type and may be very different to the value

of 2.65 g/cm3 in Linn and Doran, 1984? Please give uncertainty for the WFPS.

**Response:** In Linn and Doran (1984), they assumed the particle density of mineral soils to be 2.65 g/cm$^3$.

We admitted that soil particle density would vary with soil type, but the soil in our experimental field was

made of 50% silt, 25% sand, and 25% clay (Zhang et al., 2016), which is a kind of mineral soil. Therefore,

525 we still use 2.65 g/cm$^3$ as the soil particle density in the revised manuscript and added a short sentence to

define it as "soil porosity = (1 – soil bulk density / 2.65), assuming a particle density of 2.65 g cm$^{-3}$, which

is the most commonly used value for mineral soils (Linn and Doran, 1984)."

Line 199-200: For the typical applied extraction by unbuffered KCl solution underestimation of nitrite is

530 well-known, see Homyak et al., 2015 (https://doi.org/10.2136/sssaj2015.02.0061n). Here the nitrite

concentrations should be considered as lower limit.

**Response:** We admit that this method will underestimate soil nitrite concentrations (Miyasaka et al.,

2014), but we did not measure the concentration of nitrite but the concentration of ammonium and nitrate.

And, this method is reliable in determining the concentration of ammonium and nitrate in soil (Keeney

535 and Nelson, 1983). Therefore, we have not made any changes in this paragraph.

Line 214: Better give an average daytime J(NO2)? The total campaign average will be otherwise restricted by the zero nighttime values.

**Response:** Thank you for your advice. We provided an average daytime value of $J(NO_2)$ to replace the total campaign averaged value as "Except for the value less than $1.0 \times 10^{-4}$ s$^{-1}$, the average $J(NO_2)$ during the campaign was $3.12 \times 10^{-3}$ s$^{-1}$."

Line 236-239: Fluxes measured in other studies are not only similar in comparison to the CK plots but also to all (CK and NP) PFP and LEP data. I.e. only the fluxes observed shortly after fertilization (HEP) with extremely high N-amount are much higher (see major concern 2).

**Response:** These sentences were revised and moved to the end of the next paragraph as follows:

"Except the data from NP plots during HEP, the other fluxes (including fluxes from CK plots during all periods and fluxes from NP plots during PFP and LEP) are similar to observations at other sites with no nitrogen fertilization application, such as grass (von der Heyden et al., 2022) or forest (Ramsay et al., 2018; Sörgel et al., 2015; Zhou et al., 2011) regions."

Line 273: add references of Meng et al., 2022 and von der Heyden et al., 2022 with similar conclusions.

**Response:** Added as suggested.

Line 277-280: First, also in Ren et al. (CALNEX data) and Laufs et al. soils were studied which are regularly fertilized. I.e. they are not "non-fertilized". Second, the fluxes from the two studies are also in agreement with the present study, if the normal PFP and LEP data - even of fertilized soils - are considered. Thus, these low fluxes represent more typical conditions, while only during the short HEP period (few weeks) extremely high fluxes (two orders of magnitude higher) are observed for "over-fertilized" soils, see major concerns.

**Response:** Thank you for your meaningful suggestion. We revised the description of the fertilization condition in Laufs et al. (2017) and Ren et al. (2011) as "Laufs et al. (2017) and Ren et al. (2011) also conducted flux measurements in farmland with regular fertilized and they found high correlations between HONO flux and the product of $NO_2$ concentrations and $J(NO_2)$ or solar radiation."

Besides, we also agree that high emissions only appeared a few days after fertilization and highlighted this in Section 3.2 as "Overall, except the data from NP plots during HEP, the other fluxes (including fluxes from CK plots during all periods and fluxes from NP plots during PFP and LEP) are similar to observations at other sites with no nitrogen fertilization application, such as grass (von der Heyden et al., 2022) or forest (Ramsay et al., 2018; Sörgel et al., 2015; Zhou et al., 2011) regions."

Line 290-293: Delete the references by Laufs et al, 2017; Meng et al, 2022; Ren et al., 2011; Sörgel et al., 2015; Zhou et al., 2011, since in these studied two orders of magnitude lower fluxes were determined.

**Response:** These references were deleted as suggested in the revised manuscript.

Line 320: exchange order of NO2-/ H+ (acid-base…)

**Response:** This sentence has been modified as (please point it out if we misunderstood this comment):

Soil $NO_2^-$ (aq) can combine with $H^+$ (aq) and form $HNO_2$ (aq), which is an acid-base process. $HNO_2$ (aq) can be released into the atmosphere as HONO through liquid-gas partitioning.

Section 3.5.2: If the temperature and humidity are the driving forces of the soil emissions, than the shape of the diurnal HONO flux profile should be similar with that of the temperature (and the inverse of the relative humidity). However, in most studies the HONO fluxes maximize around noon (see also Fig. 5A of the present study…), while the temperature shows a maximum in the afternoon (heat capacity of the soil, when heated up by the sun during daytime). Thus, at least under "normal" conditions, there must be other explanations for the HONO fluxes. Here I recommend that the authors also show a correlation of the normal diurnal flux data (Figure 5A) with the product of J(NO2) x [NO2] as done in other studies. I expect that the correlation is much better compared when using the temperature.

Only for extreme HEP conditions (see Figure 5B, representative for a few weeks per year for "over-fertilized" soils…) the diurnal profile of F(HONO) may well follow that of the temperature (see also the correlation results shown in Fig. 6 for HEP).

**Response:**

As we described in the Method, samplings at 2h internals were only performed during PFP and HEP periods from the NP plot. And for the PFP period, the HONO fluxes were only available for 3–4 days

(13–17, June). Unfortunately, during these days, except for the period from 7:00 to 23:00 on June 15, the data on HONO flux and atmospheric $NO_2$ were not available simultaneously. Therefore, we made a correlation analysis between the HONO flux and the product $[NO_2] \times J(NO_2)$ during this period and found that the correlation ($R^2 = 0.89$) was indeed significantly greater than the correlation between the HONO flux and air temperature. However, during the HEP period, the correlation was significantly worse ($R^2 = 0.43$). The figures exhibit the correlation between the HONO flux and the product $[NO_2] \times J(NO_2)$ were added to the Supplement Materials and showed as follows:

[Figure]

Figure S3. Correlation of the HONO flux with the product $[NO_2] \times J(NO_2)$ during 07:00 to 23:00 UTC, June 16, 2021 (A) and HEP (B). HEP: high HONO emission period.

At the end of Section 3.3, we also add some sentences to discuss the contribution of photosensitized heterogeneous reactions of $NO_2$ on the soil surfaces to measured soil HONO emissions during different periods as:

Laufs et al. (2017) and Ren et al. (2011) also conducted flux measurements in farmland, and they found high correlations between HONO flux and the product of $NO_2$ concentrations ($[NO_2]$) and $J(NO_2)$ or solar radiation. A similar good correlation was found during the PFP period of our present study (Fig. S3A, $R^2 = 0.89$). These findings suggest that photosensitized heterogeneous reactions of $NO_2$ on the soil surfaces may be the main sources of the observed HONO flux under typical non-fertilization conditions. However, after fertilization application, HONO flux from the NP plot showed a weaker correlation with the product $[NO_2] \times J(NO_2)$ during HEP (Fig. S3B, $R^2 = 0.43$). Moreover, the fluxes from the CK plots (no fertilization, representative for $NO_y$-to-HONO conversion on the ground surface, see Method) are about 2 orders of

magnitude lower than those from the NP plots during HEP. Therefore, the NO$_2$ or other NO$_y$ reactions are not the main drivers of the observed HONO fluxes during HEP.

Line 357-359: This may be valid only for untypical HEP conditions. For the more typical PFP and LEP conditions, I expect a photochemical origin (s. above: check by plotting F(HONO) against J(NO2) x NO2).

**Response:** Yes. Please see the response to the above comment.

Section 3.6: Re-evaluate the whole section after considering the nonlinear relation between F(HONO) and fertilization amount.

**Response:** We re-estimated the regional HONO emissions after considering the exponential relation between HONO emission factors and fertilization application rates. The related discussion was already shown above.

Conclusion 1): Add a conclusion that the average F(HONO) data for all PFP and LEP conditions are in excellent agreement with most flux studies under normal fertilization conditions.

**Response:** We added your suggestion to conclusion 1) in the revised manuscript as follows:

"However, as for other conditions (non-fertilized or long times after fertilization), the averaged HONO fluxes are in excellent agreement with most flux studies with small fertilizer application rates."

We didn't use the word "normal fertilization" because the high FAR is not only used for the experimental agricultural fields in this study but also for vast agricultural fields in China and other countries. So, this is not a special case but might be a "common" one in those regions. Nevertheless, we agree that reducing fertilizer application is a potential air quality mitigating strategy. See more details in the response to Major Comment-1.

Conclusion 2): Add that these results only hold for extremely high nitrogen application during short periods after fertilization in China.

**Response:** Conclusion 2) was improved as: "Soil is an important and even dominant source of daytime HONO during short periods after intensive fertilizer application events (higher FARs). The observed HONO flux after fertilization can explain daytime HONO missing sources previously reported at this site

and in other rural regions. Therefore, the synchronous measurements of fluxes and ambient concentrations are crucial to understanding the HONO budget as well as the follow-up atmospheric impacts on air quality, e.g., the regional abundance of $O_3$ and aerosol. Moreover, we found that deep-burying fertilizer can reduce soil HONO emissions compared to traditional spreading on the soil surface, constituting a potential HONO emission reduction strategy."

Conclusion 3): please do not use a constant value of EF(HONO), see above.

**Response:** Thanks. EF calculations have been significantly improved. See response to Major Comment-2.

References:

General:

Order the references with same first authors chronologically (i.e. increasing year of publication), see VandenBoer et al., Wang (Y.) et al., Wu et al., Xue et al.. In addition, if an author has two publications in one year (a and b) use first the reference a) in the text.

**Response:** Changed as suggested.

Line 500, 580, 640, 716, 731: Pöschl,

**Response:** Revised.

Line 522, 688: Dubé

**Response:** Revised.

Line 526: Lörzer

**Response:** Revised.

Line 528: the doi link is not working?

**Response:** Revised.

Line 533-534 and others (644-647, 650-653, 660-661, 708-711, 723, 729, 780): please unify the style for Chinese given names and use only the first letter. E.g. Line 533: should be Li, D. and not Li, D. D.!

**Response:** Improved as suggested.

Line 537, 543, 645: Häseler

**Response:** Revised.

Line 544: Jäger

**Response:** Revised.

Line 557 and 594: Min, K.-E.

**Response:** Revised.

Line 579: …-Röser

**Response:** Revised.

Line 579, 715, 731: Sörgel

**Response:** Revised.

Line 654: subscript the "x" in ROx

**Response:** Revised.

Line 688: Öztürk

**Response:** Revised.

Line 703: 57(9), 3516-3526 missing

**Response:** Revised.

Line 710: Petäjä

**Response:** Revised.

Line 723: Huang, X.-R. Y.

**Response:** Revised.

Line 727: paper number e2021JD036379 missing

**Response:** Revised.

Line 729: Müller

**Response:** Revised.

Line 730: Fröhlich-…

**Response:** Revised.

Line 767: HOx and subscript the "x"

**Response:** Revised.

Line 772: use paper number L15820 (delete n/a-n/a)

**Response:** Revised.

Table 1: Here max fluxes are compared with the maximum in the average diurnal flux profile (see footnote d). I recommend to use the latter also for all other studies, since outliers are removed by this approach. Even better, one could simply compare average fluxes. In this case in all studies (except the OTDC fresh fertilization studies) the average fluxes would be in the range 0.1-2 ng N/m2/s…

**Response:** Therefore, we summarized the average and maximum of flux measurements and the average diurnal profile.

Table 1. Summary of the maximum values of HONO flux in field measurements over different soil types and corresponding measurement methods and fertilizer application rates (FAR) worldwide.

| Soil type | Method | HONO flux (ng N m$^{-2}$ s$^{-1}$) | FAR (kg N ha$^{-1}$) | References |
| --- | --- | --- | --- | --- |

| | | Mean | Max-1[a] | Max-2[b] | | |
|---|---|---|---|---|---|---|
| Agriculture | REA[c] | - | 7.0 | 1.4 | 0 | 1 |
| Forest | REA | 1.4 | 18.3 | 2.7 | 0 | 2 |
| Grassland | REA | - | 2.3 | 1.0 | 0 | 3 |
| Forest | AG[d] | 0.56 | 0.98 | - | 0 | 4 |
| Maize | AG | - | - | 2.3 | 33.4 | 5 |
| Wheat | AG | 0.84 | 15.4 | 2.8 | 69 | 6 |
| Maize | OTDC[e] | - | 1515 | - | 330 | 7 |
| Maize | OTDC | 21 | 40 | - | 45 | 7 |
| Maize | OTDC | - | 40 | 20 | 180 | 8 |
| Wheat | OTDC | 2.9 | 7.7 | 5.7 | 69 | 9 |
| Agriculture | OTDC | 34 | 348 | 83 | 247 | 10 |
| Maize | OTDC | 63 | 372 | 126 | 300 | This study |

730 [a]: maximum values in the time series; [b]: maximum values in the diurnal variations; [c]: relaxed eddy accumulation; [d]: aerodynamic gradient; [e]: open-top dynamic chamber.

1: (Ren et al., 2011); 2: (Zhou et al., 2011); 3: (von der Heyden et al., 2022); 4: (Sörgel et al., 2015); 5: (Laufs et al., 2017); 6: (Meng et al., 2022); 7: (Xue et al., 2019); 8: (Tang et al., 2019); 9: (Tang et al., 2020); 10: (Xue et al., 2022).

735

Fig. 1: In Figure 1(B) I do not see items 10. and 11.? Please add to the figure. In addition, please specify in the methods section which type of pump (5.) is used (Teflon membrane pump?). Here additional artifacts may appear depending on the type of pump used, see major issue 1a)…

**Response:** Sorry for misnumbering these items. And we re-numbered all devices in the revised
740 manuscript. Pump (5.) is a diaphragm pump, which affects little flush gases. We added the type of the diaphragm pump in the methods section as follows:

"As shown in Figure 1, each group contains three replicated Exp-chambers and one Ref-chamber that are flushed by the same air pumped by diaphragm pumps (N838KNE, Germany) from the top of the metal sample tube (I.D. of 4 cm, 2 m in height, with the inner wall coated with Teflon film)."

745

Figure 2. Here a very strong rain event (ca. 150 mm) is shown around the 9. September, while J(NO2) is very high at that day? In contrast, for the 19. September there is almost no light intensity (J(NO2)) but no rain? Please check the rain data. Here I find different rain periods in Figure S2? In addition, is there no J(NO2) data in the period 11-15. of July, or is it very low?

750 **Response:** Thanks for pointing out this mistake. It is an issue associated with plotting. We found that the ordinate setting of rainfall data in Figure 2(B) was wrong, and we corrected it in the revised manuscript. The new Figure 2 is as follows:

[Figure]

Figure 2. The time series of meteorology (A: air temperature and relative humidity; B: air pressure and rainfall; C: the
755 photolysis frequency of $NO_2$ (J($NO_2$)) during maize season at the experiment site.

In order to protect the J($NO_2$) sensors, they were shut down during heavy rain (with a cumulative rainfall of more than 100 mm) period from 11 to 15 July. Therefore, there was no data during this period.

Figure 3: Unfortunately, after the HEP period there is a data gap, after which the HONO fluxes are again
760 very low in the LEP period. However, if I would fit any Gaussian profile into the HEP flux data, I would get much higher HONO fluxes in the early LEP period. Was the soil anyhow treated directly after the HEP period? How do you explain the sudden step in the flux profile? Explain the terms NP and CK in the figure caption (a figure should stand alone).

**Response:** As shown in Figure 2, there was heavy rains (with a cumulative rainfall of more than 100 mm) after the HEP period. Then, HONO was hardly emitted from soils. Therefore, we could not fit the flux data into these days by any Gaussian profile, which must significantly overestimate HONO fluxes. In the revised manuscript, we added a sentence to explain the sudden step after the HEP period:

"A heavy rainstorm (with a cumulative rainfall of more than 100 mm) fell during 11–13 July, significantly limiting HONO release from the soil. During LEP, the average $F_{HONO-NP}$ and $F_{HONO-CK}$ were 1.03 and -2.88 ng N m$^{-2}$ s$^{-1}$, respectively."

Besides, the terms NP and CK were explained in all figure captions in the revised manuscript.

Figure 4/5/6: dito for PFP, HEP and NP in figure caption 4/5/6.

**Response:** Changed as suggested.

Figure 6: Can you also show a plot against J(NO2) x [NO2]? Could be also in the supplement.

**Response:** We added a plot showing the correlation between HONO flux with the product $[NO_2] \times J(NO_2)$ during PFP and HEP into the Supplementary Material, which has been shown in the former response. The results have also been discussed in Section 3.3, as mentioned above.

Figure 8: Show a new figure considering the nonlinear relation between F(HONO) and fertilization amount.

**Response:** Yes, we have considered the exponential relationship. See response to Major Comment-2.

Figure S3: Describe the arrows in the lower figure in the caption. E.g. "The arrows indicate reduced HONO fluxes after rain events."

**Response:** The description was added in the revised Supplementary Material.

---

## Author Comment (AC2)

[revised manuscript text omitted]

LEP. Shadows represent half of the standard deviation (±0.5 σ).

[Figure]

**Figure 7. (A): the national fertilizer application amount (CF: compound fertilizer; KF: potash fertilizer; PF: phosphatic fertilizer; NF: nitrogen fertilizer) in China during 1978–2021 and applied nitrogen amount (NF amount + 0.22 × CF amount) in the North China Plain during 1987–2021; (B): the annual HONO emissions from fertilized fields in 2021 in China (HONO emission factors changed exponentially with fertilizer application rate in each province). Data source: China Statistical Yearbooks 1979–2022.**

[Figure]

**Figure 8. The diurnal variations of $O_3$, $NO_2$, and $O_X$ ($O_3$ + $NO_2$) concentrations during PFP, HEP, and LFP. Shadows represent half of the standard deviation ($\pm 0.5$ σ). PFP: pre-fertilization period; HEP: high HONO emission period; LEP: low HONO emission period.**

[Figure]

**Figure 8. (A): the national fertilizer application amount (CF: compound fertilizer; KF: potash fertilizer; PF: phosphatic fertilizer; NF: nitrogen fertilizer) in China during 1978–2020 and applied nitrogen amount (nitrogen fertilizer amount + 0.25 × compound fertilizer amount) in the North China Plain during 1987–2020; Data source: China Statistical Yearbooks 1979–2021. (B): the annual HONO emissions from fertilized fields in 2020 in China (0.68% of applied nitrogen was lost via HONO).**